# Resp-Agent: An Agent-Based System for Multimodal Respiratory Sound Generation and Disease Diagnosis

**Pengfei Zhang, Tianxin Xie, Minghao Yang, Li Liu***
The Hong Kong University of Science and Technology (Guangzhou)

## Abstract

Deep learning-based respiratory auscultation is currently hindered by two fundamental challenges: (i) inherent information loss, as converting signals into spectrograms discards transient acoustic events and clinical context; (ii) limited data availability, exacerbated by severe class imbalance. To bridge these gaps, we present **Resp-Agent**, an autonomous multimodal system orchestrated by a novel Active Adversarial Curriculum Agent (Thinker-$A^2$CA). Unlike static pipelines, Thinker-$A^2$CA serves as a central controller that actively identifies diagnostic weaknesses and schedules targeted synthesis in a closed loop. To address the representation gap, we introduce a modality-weaving Diagnoser that weaves clinical text with audio tokens via strategic global attention and sparse audio anchors, capturing both long-range clinical context and millisecond-level transients. To address the data gap, we design a flow matching Generator that adapts a text-only Large Language Model (LLM) via modality injection, decoupling pathological content from acoustic style to synthesize hard-to-diagnose samples. As a foundation for this work, we introduce **Resp-229k**, a benchmark corpus of 229k recordings paired with LLM-distilled clinical narratives. Extensive experiments demonstrate that Resp-Agent consistently outperforms prior approaches across diverse evaluation settings, improving diagnostic robustness under data scarcity and long-tailed class imbalance. Our code and data are available at `https://github.com/zpforlove/Resp-Agent`.

## 1 Introduction

Respiratory auscultation is a fundamental component of clinical diagnosis, providing critical acoustic evidence for assessing pulmonary health (Heitmann et al., 2023; Bohadana et al., 2014). Accurate and automated analysis of respiratory sounds holds substantial clinical value for the early screening, diagnosis, and monitoring of respiratory diseases (Rocha et al., 2019). Although deep learning has driven significant progress in this domain, existing methods remain constrained by fundamental limitations that hinder both performance and practical deployment (Huang et al., 2023; Xia et al., 2022; Coppock et al., 2024).

The first challenge is a unimodal representational bottleneck. Audio models often convert signals into mel-spectrograms for image-style CNNs (Bae et al., 2023; He et al., 2024), which discards phase and blurs fine temporal structure, obscuring transient events such as crackles (Paliwal et al., 2011). Conversely, text-only models capture electronic health record (EHR) context but lack objective acoustic evidence, limiting discrimination between conditions with similar narratives but distinct auscultatory patterns. Notably, large-scale health acoustic pretraining (Baur et al., 2024) remains unimodal; without deep multimodal fusion, performance and reliability saturate.

The second limitation is the lack of large, well-annotated multimodal datasets. Most public respiratory-sound corpora are small, cover only a few conditions, and lack systematic curation (Zhang et al., 2024a). Even when auxiliary metadata such as demographics and symptoms is available, existing approaches rely on basic fusion techniques and task-specific designs, limiting the development of generalized multimodal models (Zhang et al., 2024b).

---

*Corresponding author: avrillliu@hkust-gz.edu.cn

A third challenge lies in the disconnect between analysis and generation. Current research is heavily skewed towards diagnostic tasks like classification and detection (Huang et al., 2023; Xia et al., 2022), leaving the potential of generative modeling largely unexplored (Kim et al., 2023). The ability to synthesize respiratory sounds with specific pathological characteristics would not only support medical education, data augmentation, and interpretability research but also serve as a stringent test of a system's multimodal understanding. However, no existing framework unifies analysis and synthesis within a single coherent system (Zhang et al., 2024a;b).

To systematically address these challenges, we introduce Resp-Agent, a multimodal agent framework inspired by the design philosophy of intelligent agents. Resp-Agent decomposes complex functionality into specialized modules coordinated by a central controller that plans and schedules tasks. This design enables unified processing of respiratory sounds for both diagnostic analysis and generative synthesis, advancing the state of multimodal respiratory intelligence.

In summary, we propose **Resp-Agent**, a closed-loop framework that turns passive analysis into *generation ↔ diagnosis* co-design. The contributions of our method are:

1) **Resp-229k: A large-scale, clinically contextualized benchmark.** Resp-229k contains approximately 408 hours and 229k respiratory recordings spanning 16 diagnostic categories. We pair each sample with a clinical narrative synthesized from EHR records using LLMs and refined for accuracy. The benchmark features source-disjoint splits to rigorously test model generalization, while the correspondence between text and audio supports multimodal modeling and transparent verification.

2) **Controllable Synthesis.** We design a Generator that augments a compact LLM to synthesize high-fidelity respiratory audio. Disease semantics are conditioned on text, while acoustic style is captured by BEATs(Chen et al., 2023) tokens. A conditional flow-matching decoder reconstructs waveforms with high fidelity. This design is instantiated as RESP-MLLM, to the best of our knowledge, the first multimodal large language model trained with aligned text–audio supervision for controllable respiratory sound synthesis. Flow matching ensures stable, phase-aware reconstruction of transient events, and the BEATs-derived style tokens, which model device and timbre factors, are essential for clinical realism.

3) **Robust Diagnosis.** We introduce a Diagnoser based on *modality weaving*, which interleaves audio embeddings with text. A strategically designed global-attention mechanism enables the model to jointly condition on fused text while parsing the acoustic stream at ≈80ms resolution, capturing fleeting events that characterize respiratory sounds. By leveraging a Longformer(Beltagy et al., 2020) backbone to capture long-range dependencies, our approach yields superior performance and improved generalization across varying domains.

## 2 RELATED WORK

Respiratory Sound Classification (RSC) has largely relied on audio-only pipelines trained on small, single-source datasets, which limits out-of-domain generalization. Standard approaches utilize pretrained backbones like PANNs (Kong et al., 2020) and AST (Gong et al., 2021) to mitigate data scarcity. The OPERA benchmark has begun to close the data gap via domain-specific pretraining (Zhang et al., 2024a); however, most systems remain constrained by single-modality supervision and in-distribution evaluation. Our work departs from this paradigm in two key ways. **(i) Multimodal fusion.** We weave EHR-style textual tokens and acoustic tokens within a long-context Transformer. Unlike RespLLM (Zhang et al., 2024b), which feeds concatenated modality tokens through dense full attention, we introduce *Strategic Global Attention* with sparse audio anchors, drawing on ideas from efficient Transformers (Beltagy et al., 2020; Zaheer et al., 2020) to route clinical context to transient acoustic events at sub-quadratic cost. We evaluate on source-disjoint splits to stress-test generalization under realistic distribution shifts (Koh et al., 2021) and label imbalance (Johnson & Khoshgoftaar, 2019). **(ii) Targeted augmentation.** Recent generative models enable high-fidelity audio synthesis (Borsos et al., 2023; Lipman et al., 2023; Liu et al., 2023; Peebles & Xie, 2023), but existing augmentation strategies are typically untargeted, relying on generic perturbations such as SpecAugment (Park et al., 2019) or unconditional generation (Kim et al., 2023). We introduce **Resp-Agent**, a closed-loop system in which an LLM-based Thinker-A$^2$CA diagnoses model failures and requests condition-controlled synthesis from a flow-matched generator, turning augmentation into a precise instrument for adversarial edge-case creation and distribution balancing.

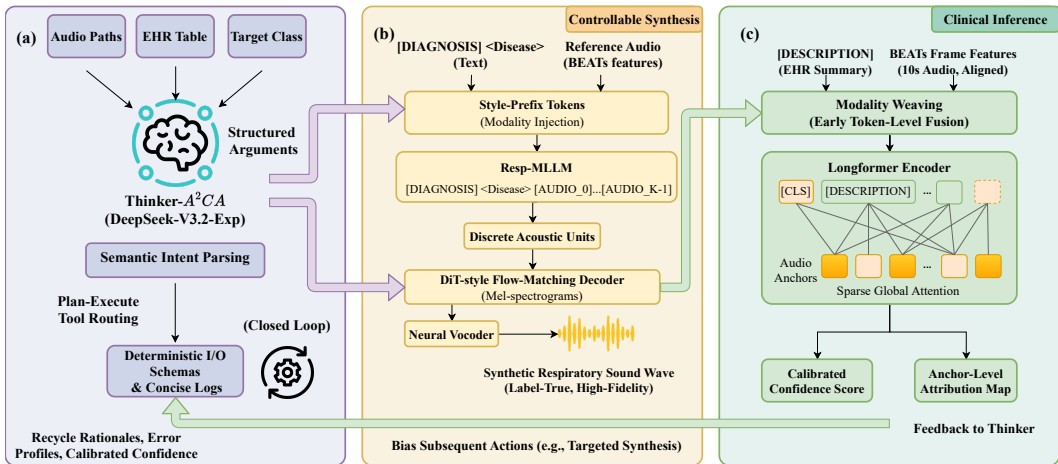

Figure 1: Overview of Resp-Agent. The framework functions as a closed-loop system composed of three interacting modules: **(a) Thinker:** A compute-aware planner (Thinker-$A^2CA$) that parses semantic intents and routes tasks to other agents based on recycled error profiles and calibrated confidence. **(b) Generator:** A synthesis module utilizing *modality injection* to condition the Resp-MLLM on both textual diagnosis and reference acoustic style, decoding discrete units via conditional flow matching. **(c) Diagnoser:** A clinical inference module employing *modality weaving* to fuse EHR summaries with audio features early in the network, leveraging sparse global attention for robust cross-modal reasoning.

# 3 RESP-229K: A LARGE-SCALE, MULTI-SOURCE, CROSS-DOMAIN BENCHMARK

We introduce **Resp-229k** to address the scarcity of multimodal supervision and the lack of robust cross-domain evaluation in respiratory sound analysis. Unlike existing datasets, Resp-229k provides paired audio with standardized clinical summaries, converting diverse metadata into a format suitable for multimodal modeling. We also establish a strict out-of-domain evaluation protocol to explicitly test model generalization. We aggregate 238,074 raw clips from five public databases; after discarding corrupted or too-short recordings, 229,101 quality-controlled samples remain, spanning 16 classes (15 conditions and 1 control), and are used in all main-paper experiments.

A core contribution is the textual supervision. Instead of full electronic health records, each clip is paired with a standardized clinical summary, a concise paragraph synthesized from available source fields. Summaries adapt to source coverage: when demographics and symptoms exist, they are included; when only auscultation events and acquisition context are present, the summary focuses on those. Concretely, we retain two typical regimes as a modeling challenge: technical/event-driven summaries (auscultatory events, site, sensor/filter, phases, wheezes/crackles) and clinically enriched summaries (demographics, smoking status, comorbidities, symptoms, past medical history).

We programmatically convert heterogeneous CSV/TXT/JSON fields and filename-derived codes into standardized summaries using DeepSeek-R1-Distill-Qwen-7B (Guo et al., 2025) as a lightweight data-to-text engine. The model does not interpret audio; instead, it consolidates existing metadata into a schema-grounded paragraph with a consistent style across sources, enabling reproducible, low-cost annotation refreshes while preserving diagnostically relevant heterogeneity.

To mitigate hallucination and governance risks, all LLM-generated clinical summaries undergo a second-stage audit that combines rule-based consistency checks, critique from a stronger reasoning model acting as a verifier, and sampling-based human review. This process ensures that only summaries that pass the pipeline, or are rewritten and reverified after being flagged, are retained in Resp-229k. A detailed description of the auditing pipeline is provided in Appendix E.

To standardize comparisons, we specify two tasks and metrics: (i) multimodal disease classification, reporting accuracy and macro-F1; and (ii) controllable audio generation conditioned on disease semantics, reporting objective acoustic similarity and clinical-event fidelity. We report both in-domain validation results and strictly out-of-domain test results. For evaluation, Resp-229k enforces

Table 1: Resp-229k overview: split statistics and source datasets. The dataset identifiers correspond to UK COVID-19 (Coppock et al., 2024; Budd et al., 2024; Pigoli et al., 2022), ICBHI (Rocha et al., 2017), SPRSound (Zhang et al., 2022), COUGHVID (Orlandic et al., 2021), and KAUH (Fraiwan et al., 2021).

**(a) Resp-229k split statistics (effective samples)**

| Split | #Files | Hours | Mean (s) | Max (s) |
|---|---|---|---|---|
| Train | 196,654 | 341 | 6.2 | 86 |
| Valid | 16,931 | 31 | 6.6 | 71 |
| Test | 15,516 | 36 | 8.4 | 30 |
| Total | 229,101 | 408 | 6.4 | 86 |

**(b) Source datasets for the curation of Resp-229k**

| Name | Role | Device | Sample Rate (kHz) | Mean Duration (s) |
|---|---|---|---|---|
| UK COVID-19 | Train/Validation | Microphone | 48 | 5.9 |
| ICBHI | Train/Validation | Stethoscope | 4–44.1 | 22.2 |
| SPRSound | Train/Validation | Stethoscope | 8 | 11.0 |
| COUGHVID | Test | Microphone | 48 | 6.9 |
| KAUH | Test | Stethoscope | 4 | 15.0 |

a strict cross-domain split: training/validation on ICBHI, SPRSound, and UK COVID-19, and testing exclusively on KAUH and COUGHVID (unseen during training). This design assesses robustness across institutions, sensors, and collection protocols. Concise split statistics and per-dataset metadata appear in Table 1. A complete specification of the 16-class label space (15 disease categories plus a healthy control group) is provided in Appendix A.

## 4 RESP-AGENT: AN LLM-ORCHESTRATED LOOP FOR UNIFIED DIAGNOSIS AND CONTROLLABLE SYNTHESIS

The overall architecture of Resp-Agent is depicted in Figure 1. Given the paired text–audio supervision and the cross-domain split established by Resp-229k, Resp-Agent is designed as a centrally planned, compute-aware multi-agent system that integrates standalone audio and NLP modules into a closed loop. A compute-efficient planner, Thinker-A$^2$CA (DeepSeek-V3.2-Exp; (Liu et al., 2025)), performs semantic intent parsing and plan–execute tool routing using structured arguments (audio paths, EHR tables, and target classes), enforcing deterministic I/O schemas and emitting concise, instrumented logs. Beyond dispatch, the controller reuses model rationales, error profiles, and calibrated confidence to bias subsequent actions (e.g., targeted synthesis for failure modes), thereby coupling data generation and diagnosis under tight accelerator budgets without compromising coverage or reproducibility. The Thinker coordinates two task-specific agents detailed below: a **Generator** (Section 4.1) that synthesizes controllable respiratory audio via modality injection and conditional flow matching, and a **Diagnoser** (Section 4.2) that fuses clinical narratives with audio tokens through modality weaving and strategic global attention.

### 4.1 GENERATOR: DISCRETE-UNIT PLANNING AND CFM RECONSTRUCTION

We target controllable respiratory-sound synthesis that disentangles pathological content (what to generate) from timbral style (how it should sound). The Generator follows a two-stage design. Stage 1 retools a unimodal LLM into a multimodal unit generator conditioned on diagnosis semantics and a reference style, as illustrated in Figure 2. Stage 2 reconstructs high-fidelity audio from the predicted discrete units via conditional flow matching (CFM) and a neural vocoder.

#### 4.1.1 STYLE-CONDITIONED UNIT MODELING WITH A RETOOLED LLM

We retool a light text-only backbone (Qwen3-0.6B-Base(Yang et al., 2025)) into a truly multimodal unit generator and denote the trained model Resp-MLLM. The conversion relies on modality injection with a trainable style projector while leaving the language backbone architecture intact. Let $\mathbf{Z} \in \mathbb{R}^{T \times D}$

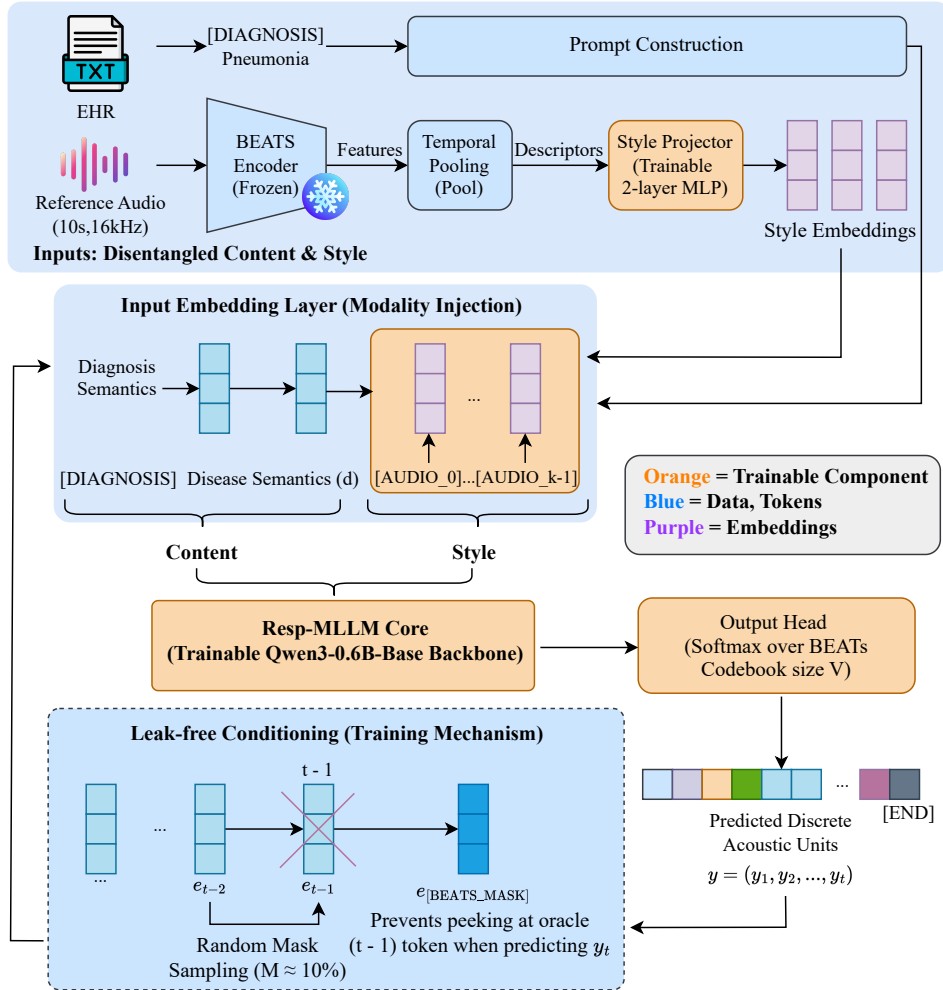

Figure 2: Detailed architecture of Resp-MLLM (Stage 1 of the Generator). The model functions as a style-conditioned multimodal unit generator. **Top:** A *modality injection* mechanism fuses textual diagnosis semantics with acoustic style embeddings (projected from temporally pooled BEATs features) to prompt the Qwen3-0.6B-Base backbone. **Bottom:** A *leak-free conditioning* strategy is employed during training: random mask sampling ($\mathcal{M} \approx 10\%$) prevents the model from peeking at oracle tokens, ensuring robust autoregressive prediction of discrete acoustic units.

be framewise BEATs features from a $10\,\mathrm{s}$, $16\,\mathrm{kHz}$ reference ($T{=}496$). We compress $\mathbf{Z}$ into $K$ style descriptors and map them to the LLM hidden space via a two-layer MLP:

$$\mathbf{P} = \mathrm{Pool}_K(\mathbf{Z}) \in \mathbb{R}^{K \times D}, \qquad \mathbf{E}^{\mathrm{style}} = \mathrm{StyleProj}(\mathbf{P}) \in \mathbb{R}^{K \times H}. \qquad (1)$$

In the input, we reserve $K$ placeholders $[\texttt{AUDIO}_0], \ldots, [\texttt{AUDIO}_{K-1}]$ and replace their embeddings with rows of $\mathbf{E}^{\mathrm{style}}$. The mixed prompt

$$\underbrace{[\texttt{DIAGNOSIS}]\, d}_{\text{content}}\ \underbrace{[\texttt{AUDIO}_0] \cdots [\texttt{AUDIO}_{K-1}]}_{\text{style}}$$

binds disease semantics $d$ to a reference timbre without modifying the language stack. Resp-MLLM then autoregressively predicts a sequence of discrete acoustic units $\mathbf{y} = (y_1, \ldots, y_L)$ from a BEATs codebook of size $V$:

$$\mathcal{L}_{\mathrm{Resp}} = -\sum_{i=1}^{L} \log p_{\mathrm{Resp}}\big(y_i \mid y_{<i}, d, \mathbf{E}^{\mathrm{style}}\big),$$

$$\text{s.t. } y_i \in \{0, \ldots, V-1\}. \qquad (2)$$

To avoid inadvertent teacher forcing while preserving causal decoding, we apply a lightweight masked-input scheme. Let $\mathcal{T}$ index the target unit segment and $\mathcal{M} \subset \mathcal{T}$ be a random subset (e.g., $\approx 10\%$). For each $t \in \mathcal{M}$ we replace the preceding input embedding by a dedicated vector $\mathbf{e}_{\texttt{[BEATs\_MASK]}}$:

$$\mathbf{e}_{t-1} \leftarrow \mathbf{e}_{\texttt{[BEATs\_MASK]}}, \qquad t \in \mathcal{M}, \tag{3}$$

so the model cannot peek the oracle $(t-1)$ token when predicting $y_t$. Sequences terminate with `[END]`, and padding positions are excluded from the loss. This keeps the content–style interface clean and stabilizes training of Resp-MLLM.

### 4.1.2 CONDITIONAL FLOW MATCHING FOR HIGH-FIDELITY WAVEFORMS

The predicted units serve as content for a CFM decoder parameterized by a Diffusion Transformer (DiT), which reconstructs mel-spectrograms; waveforms are then obtained using Vocos (Siuzdak, 2024). Let $\mathbf{x}_1$ be the target mel and $\mathbf{x}_0 \sim \mathcal{N}(0, \sigma^2 \mathbf{I})$ a noise prior. Flow matching learns a velocity field $v_\theta$ along the linear path $\mathbf{x}_t = (1-t)\mathbf{x}_0 + t\mathbf{x}_1$.

$$\frac{\mathrm{d}\mathbf{x}_t}{\mathrm{d}t} = v_\theta(\mathbf{x}_t, \mathbf{c}), \quad t \in [0, 1]. \tag{4}$$

The training objective minimizes the mean-squared discrepancy between the predicted and target velocities:

$$\mathcal{L}_{\mathrm{CFM}} = \mathbb{E}_{t, \, p_t(\mathbf{x}_t | \mathbf{x}_1)} \left[ \left\| v_\theta(\mathbf{x}_t, \mathbf{c}) - (\mathbf{x}_1 - \mathbf{x}_0) \right\|_2^2 \right]. \tag{5}$$

The condition $\mathbf{c}$ follows a dual-path design: (i) a content stream obtained by embedding unit indices and temporally interpolating them to the mel frame rate; and (ii) a global timbre stream formed by time-averaging BEATs features and broadcasting them across time. During training, we additionally expose a short reference prefix of the ground-truth mel in the conditioning branch (zero-padded to full length) to encourage accurate continuation while keeping the main path free-form.

This two-stage design separates what to generate (units governed by diagnosis and style prefixes) from how it should sound (timbre and continuation in CFM), enabling precise and editable control while remaining compatible with standard causal LLM training and vocoder back ends.

### 4.2 DIAGNOSER: MODALITY WEAVING WITH STRATEGIC GLOBAL ATTENTION

#### 4.2.1 INPUT-LEVEL MODALITY WEAVING

The architecture of our Diagnoser is illustrated in Figure 3. Prior work often performs late fusion by concatenating audio and text after separate encoders, which weakens alignment and underutilizes long-context transformers. We instead weave modalities at the input: EHR text tokens and a fixed audio block form a single sequence so cross-modal dependencies are modeled from the first layer.

Concretely, after tokenizing the EHR, we place a contiguous block of $T{=}496$ audio placeholders `[AUDIO_EMBED]` at a known span. Let $x$ be the waveform (16 kHz, 10 s via crop/pad), $\Phi_{\mathrm{BEATs}}(x) \in \mathbb{R}^{\tilde{T} \times D}$ the pretrained BEATs features, and $\mathrm{Align}(\cdot) : \mathbb{R}^{\tilde{T} \times D} \to \mathbb{R}^{T \times D}$ a deterministic crop/pad to $T$ steps. At the Longformer embedding layer we replace the audio placeholders in place by a learned projection:

$$\mathbf{E}_{[A]} \leftarrow \mathrm{Align}\big(\Phi_{\mathrm{BEATs}}(x)\big)\,\mathbf{W}, \qquad \mathbf{W} \in \mathbb{R}^{D \times H}, \ \mathbf{E}_{[A]} \in \mathbb{R}^{T \times H}. \tag{6}$$

Here $[A]$ denotes the audio span; all other tokens (e.g., `[CLS]`, `[DESCRIPTION]`, EHR text, `[SEP]`) use standard embeddings. This modality weaving yields a single, token-aligned stream in which audio and text interact natively. For robustness we apply light token/frame dropout to text/audio before attention (defaults $p_{\mathrm{text}}{=}0.2$, $p_{\mathrm{audio}}{=}0.1$).

#### 4.2.2 STRATEGIC GLOBAL ATTENTION

Longformer combines efficient sliding-window attention with a sparse set of global tokens. We allocate global attention to three roles: (i) the classifier `[CLS]`; (ii) a sentinel for EHR context

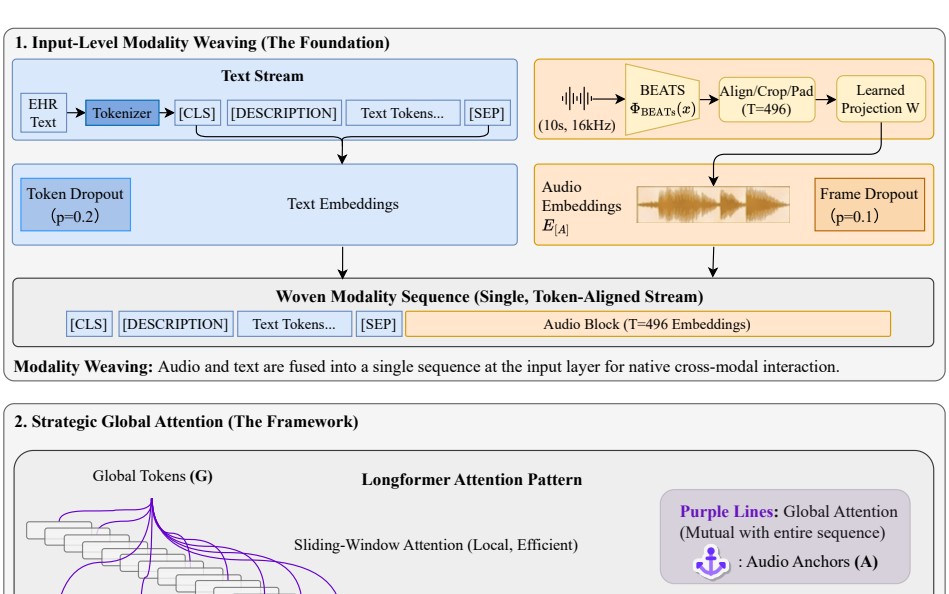

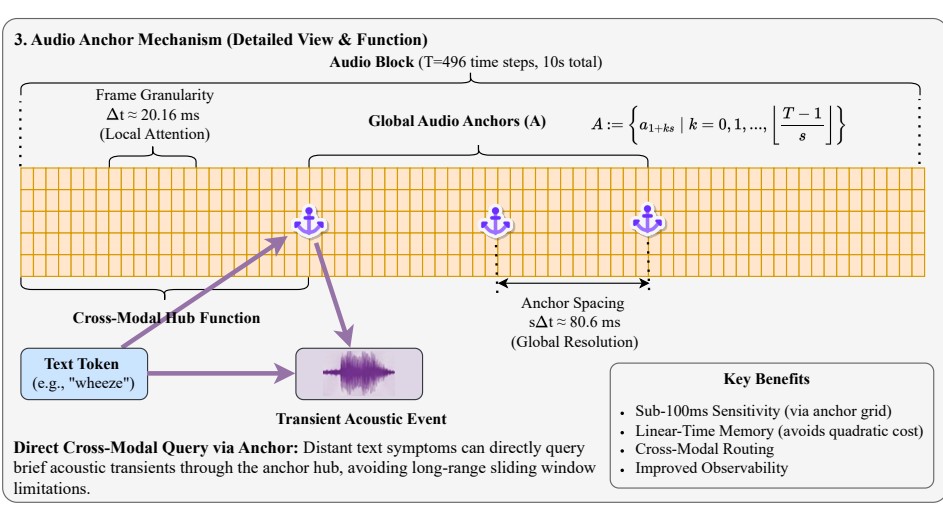

Figure 3: **Diagnoser Architecture: Modality Weaving with Strategic Global Attention.** The framework comprises three key mechanisms: **(1) Input-Level Modality Weaving:** EHR text tokens and projected audio features (extracted via BEATs) are fused into a single token-aligned sequence at the input layer, enabling native cross-modal interaction. **(2) Strategic Global Attention:** A Longformer backbone combines efficient sliding-window attention with a sparse set of global tokens, which includes textual sentinels and distributed Audio Anchors to model long-range dependencies with linear complexity. **(3) Audio Anchor Mechanism:** Anchors act as cross-modal hubs spaced at ≈80ms intervals, allowing distinct text symptoms (e.g., "wheeze") to directly query transient acoustic events, thereby capturing fine-grained temporal structures without quadratic computational costs.

[DESCRIPTION]; and (iii) stride-sampled audio "anchors" within the woven block. Let $a_1, \dots, a_T$ index audio positions and $s$ be the stride (default $s=4$). The global set is

$$
\begin{aligned}
\mathcal{A} &:= \{\, a_{1+ks} \mid k = 0, 1, \dots, \lfloor (T-1)/s \rfloor \,\}, \\
\mathcal{G} &:= \{\mathrm{pos}(\texttt{[CLS]}),\ \mathrm{pos}(\texttt{[DESCRIPTION]})\} \cup \mathcal{A}.
\end{aligned}
\tag{7}
$$

There is mutual attention between tokens in $\mathcal{G}$ and the entire sequence; all others remain local. This pattern preserves linear-time memory while creating cross-modal hubs that support long-range evidence flow. Textual symptoms (e.g., "nocturnal dry cough") can directly query transient acoustic events even when far apart.

For a $10\,\text{s}$ segment embedded into $T=496$ time steps, the per-step hop is $\Delta t = 10{,}000\,\text{ms}/496 \approx 20.1613\,\text{ms}$. With stride $s=4$, adjacent global anchors are spaced by $s\,\Delta t \approx 4 \times 20.1613 = 80.6452\,\text{ms}$. Thus, the globally queryable alignment grid affords an $\approx 80.6\,\text{ms}$ temporal resolution, with worst-case deviation $\leq s\Delta t/2 \approx 40.3\,\text{ms}$ when snapping an event to its nearest anchor. Meanwhile, local attention continues to represent features at the frame granularity $\Delta t \approx 20.16\,\text{ms}$. This strategically sparse global pattern yields sub-$100\,\text{ms}$ sensitivity to brief, low-energy respiratory phenomena (e.g., wheeze onsets, crackles) while avoiding quadratic costs. In contrast to designs that rely solely on `[CLS]` or purely local windows, anchor-based global tokens enable predictable cross-modal routing and improve the observability of rare transients, as shown in Table 8.

## 5 EXPERIMENTS

We empirically evaluate Resp-Agent on two complementary benchmarks: (i) ICBHI 4-class respiratory sound classification under the official 60–40% split, and (ii) Resp-229k, our large-scale, cross-domain, 16-class benchmark with a strict held-out test set (Test-CD). Our experiments are organized around three questions: (Q1) Does the proposed multimodal Diagnoser outperform strong unimodal and shallow-fusion baselines? (Q2) Is Thinker-guided controllable synthesis necessary beyond traditional imbalance remedies and simpler planners, and does it generalize under severe domain shift? (Q3) Are the Generator and Diagnoser architectures themselves responsible for the observed gains, rather than incidental implementation choices?

Table 2: RSC performance on the ICBHI dataset using the official 60–40% train–test split. In the Pretraining Data column, IN, AS, LA, HF, and SPR refer to ImageNet (Deng et al., 2009), AudioSet (Gemmeke et al., 2017), LAION-Audio-630K (Wu et al., 2023), HF_Lung_V1 (Hsu et al., 2021), and SPRSound, respectively. * denotes the previous state-of-the-art ICBHI Score. The **SOTA** and second-best results are highlighted in bold and by underlining, respectively.

| Method | Backbone | Pretraining Data | $S_p$ (%) | $S_e$ (%) | Score (%) |
|---|---|---|---|---|---|
| SE+SA(Yang et al., 2020) | ResNet18 | - | 81.25 | 17.84 | 49.55 |
| LungRN+NL(Ma et al., 2020) | ResNet-NL | - | 63.20 | 41.32 | 52.26 |
| RespireNet(Gairola et al., 2021) | ResNet34 | IN | 72.30 | 40.10 | 56.20 |
| Chang *et al.*(Chang et al., 2022) | CNN8-dilated | - | 69.92 | 35.85 | 52.89 |
| Ren *et al.*(Ren et al., 2022) | CNN8-Pt | - | 72.96 | 27.78 | 50.37 |
| Wang *et al.*(Wang & Wang, 2022) | ResNeSt | IN | 70.40 | 40.20 | 55.30 |
| Late-Fusion(Pham et al., 2022) | Inc-03 + VGG14 | IN | 85.60 | 30.00 | 57.30 |
| Nguyen *et al.*(Nguyen & Pernkopf, 2022) | ResNet50 | IN | 79.34 | 37.24 | 58.29 |
| Moummad *et al.*(Moummad & Farrugia, 2023) | CNN6 | AS | 75.95 | 39.15 | 57.55 |
| Bae *et al.*(Bae et al., 2023) | AST | IN+AS | 81.66 | 43.07 | 62.37 |
| Kim *et al.*(Kim et al., 2023) | AST | IN+AS | 80.72 | 42.86 | 61.79 |
| Kim *et al.*(Kim et al., 2024a) | AST | IN+AS | 79.87 | 43.55 | 61.71 |
| Kim *et al.*(Kim et al., 2024b) | AST | IN+AS | 82.47 | 40.55 | 61.51 |
| BTS (Kim et al., 2024c) | CLAP | LA | 81.40 | 45.67 | 63.54 |
| Wang *et al.*(Wang et al., 2024) | HTS-AT | IN+AS | 79.61 | 48.77 | 64.19 |
| MVST (He et al., 2024) | AST | IN+AS | 81.99 | 51.10 | 66.55 |
| Dong *et al.*(Dong et al., 2025) | AST | IN+AS | **85.99** | 49.11 | 67.55* |
| **Resp-Agent[Ours]** | LLM+Longformer | HF+SPR | 79.29 | **66.10** | **72.70** |

**Main diagnostic performance.** On ICBHI, Resp-Agent attains a score of 72.7 (Sp = 79.3, Se = 66.1), surpassing the best prior audio models by more than 5 absolute points on the official leaderboard while using a distinct LLM+Longformer backbone. This indicates that anchor-aware multimodal reasoning is competitive even against heavily pre-trained audio transformers. On Resp-229k, Table 8 in Appendix B summarizes performance under the original imbalanced and Generator-balanced regimes. The audio-only Conformer(Gulati et al., 2020) baseline reaches 0.720/0.782 Accuracy and 0.1935/0.5360 Macro-F1 (original/balanced), while the full Resp-Agent Diagnoser achieves 0.8494/0.8870 Accuracy and 0.2118/0.5980 Macro-F1. The balanced regime thus converts generative

augmentation into substantial macro-F1 gains on minority conditions without sacrificing overall accuracy.

We next fix the Diagnoser and Generator architectures and vary only the planner policy that allocates a synthetic budget $B$ over label and domain combinations on Resp-229k/Test-CD. The consolidated results are reported in Table 3.

Table 3: Summary of downstream Diagnoser performance on Test-CD under different planner policies and imbalance remedies. Macro-F1$_{tail}$ is computed over the 8 rarest classes. "–" indicates that LoSO results are not reported for that setting due to the computational cost of retraining across all five folds.

| Setting | Method | $B$ (k) | Acc | Macro-F1 | Macro-F1$_{tail}$ | LoSO Macro-F1 / tail |
|---------|--------|---------|-----|----------|-------------------|----------------------|
| | | Planner policies under matched budget (Exp. 1) | | | | |
| Test-CD | No-Synth (CE) | 0 | 0.849 | 0.212 | 0.074 | 0.237 / 0.086 |
| Test-CD | Random | 50 | 0.869 | 0.442 | 0.291 | – |
| Test-CD | Class-Prior | 50 | 0.876 | 0.512 | 0.349 | 0.473 / 0.334 |
| Test-CD | Uncertainty-Static | 50 | 0.881 | 0.546 | 0.376 | – |
| Test-CD | Thinker-A$^2$CA | 50 | **0.887** | **0.598** | **0.421** | **0.532 / 0.383** |
| | | Factorized planner variants (Exp. 2) | | | | |
| Test-CD | Rare-only | 30 | 0.873 | 0.489 | 0.381 | – |
| Test-CD | Hard-Case-only | 30 | 0.878 | 0.512 | 0.371 | – |
| Test-CD | Hard-Domain-only | 30 | 0.876 | 0.506 | 0.356 | – |
| Test-CD | Rare×Hard-Dom. | 30 | 0.881 | 0.528 | 0.397 | – |
| Test-CD | Thinker-A$^2$CA | 30 | **0.883** | **0.541** | **0.409** | – |
| | | Non-generative vs. generative imbalance remedies (Exp. 4) | | | | |
| Test-CD | CE (Baseline) | 0 | 0.849 | 0.212 | 0.074 | – |
| Test-CD | Class-Weighted CE | 0 | 0.842 | 0.248 | 0.114 | – |
| Test-CD | Focal Loss ($\gamma$=2) | 0 | 0.839 | 0.267 | 0.129 | – |
| Test-CD | CE + Class-Prior | 50 | 0.876 | 0.512 | 0.349 | – |
| Test-CD | CE + Thinker-A$^2$CA | 50 | **0.887** | **0.598** | **0.421** | – |

Experiment 1 evaluates generative planners under a matched budget of $B = 50$k synthetic clips. All planners consistently improve over the no-synthesis baseline (Macro-F1 0.212). Class-prior rebalancing reaches a Macro-F1 of 0.512, and static uncertainty sampling further improves it to 0.546, confirming that both label balancing and error-aware targeting are effective. However, the Active Adversarial Curriculum Agent (Thinker-A$^2$CA) achieves a Macro-F1 of 0.598 and a Macro-F1$_{tail}$ of 0.421, yielding sizable gains of +0.052 Macro-F1 and +0.045 Macro-F1$_{tail}$ over the strong static-uncertainty baseline at the same budget.

Experiment 2 decomposes Thinker-A$^2$CA into single-factor planners under a matched budget of $B = 30$k. Focusing only on rare labels (*Rare-only*) already lifts Macro-F1 from 0.212 to 0.489, while targeting hard cases or hard domains yields Macro-F1 scores of 0.512 and 0.506, respectively. Combining rarity and domain difficulty (*Rare×Hard-Domain*) is the strongest single-factor heuristic (Macro-F1 0.528; Macro-F1$_{tail}$ 0.397). Yet Thinker-A$^2$CA still improves further to 0.541/0.409, demonstrating that its iterative, multi-factor planning cannot be reduced to any single handcrafted heuristic.

Experiment 3 studies sample efficiency by sweeping $B \in \{0, 10\text{k}, 20\text{k}, 30\text{k}, 50\text{k}\}$. All planners exhibit diminishing returns, but Thinker-A$^2$CA is markedly more sample-efficient: at $B = 10$k, it already achieves a Macro-F1 of 0.412 ($\approx 52\%$ of its total gain over the baseline), compared with 0.378 for class-prior rebalancing and 0.331 for random sampling. This suggests that a small number of high-value, Thinker-selected clips is more beneficial than a much larger pool drawn by simpler policies.

Experiment 4 compares Thinker-guided synthesis against non-generative imbalance remedies at $B = 0$. Class-weighted cross-entropy and focal loss (Lin et al., 2017) improve Macro-F1 from 0.212 to 0.248 and 0.267, respectively, and nearly double Macro-F1$_{tail}$ (from 0.074 to 0.114/0.129). However, even the best non-generative method remains 0.331 Macro-F1 below the CE + Thinker-A$^2$CA combination at $B = 50$k (Macro-F1 0.598), indicating that the dominant source of improvement is the synthetic data itself rather than loss re-weighting.

Table 4: Compact summary of Generator content–style disentanglement

| Generator disentanglement (Exp. 6) | | | |
|---|---|---|---|
| Test | Style-Sim ↑ | P-Acc ↑ | FAD ↓ |
| Style-swap (avg over 4 styles) | 0.91 | 97.9% | 1.18 |
| Content-swap (avg over 4 labels) | 0.93 | 96.1% | 1.19 |
| Diagnoser ablations on Test-CD (Exp. 7) | | | |
| Config | Acc | Macro-F1 | |
| Late Fusion, Raw Metadata, no anchors | 0.780 | 0.145 | |
| Late Fusion, LLM EHR, no anchors | 0.790 | 0.160 | |
| Modality Weaving, Raw Metadata, no anchors | 0.640 | 0.175 | |
| Modality Weaving, LLM EHR, no anchors | 0.650 | 0.189 | |
| Modality Weaving, Raw Metadata, anchors | 0.835 | 0.195 | |
| Full Resp-Agent Diagnoser (LLM EHR + anchors) | **0.849** | **0.212** | |

Experiment 5 conducts a rigorous Leave-One-Source-Out (LoSO) evaluation across the five constituent datasets (UK COVID-19, ICBHI, SPRSound, COUGHVID, and KAUH). Averaged over folds, Macro-F1 / Macro-F1$_{tail}$ is 0.237 / 0.086 for the no-synthesis baseline, 0.473 / 0.334 for class-prior rebalancing, and 0.532 / 0.383 for Thinker-A$^2$CA (Table 3). The consistent ordering Thinker-A$^2$CA > Class-Prior > No-Synth across all held-out sources confirms that the planner's benefits are robust across sources rather than split-specific.

Experiment 6 validates that the Generator can independently control pathological content and acoustic style. In the *style-swap* setting, we fix the pathology to a rare label ("Bronchiolitis") and vary the style reference across four cross-domain clips. The Generator achieves high style similarity (Style-Sim 0.89–0.92), while the held-out Diagnoser preserves the target pathology with a Pathology-Acc of 97.5–98.1% and a Fréchet Audio Distance (FAD) of 1.14–1.21. In the complementary *content-swap* setting, a single "Control Group" style reference is held fixed while synthesizing four different pathologies. Style-Sim remains stable at $\approx$ 0.93–0.94, and Pathology-Acc remains high (94.8–97.2%) with FAD in the range of 1.17–1.22. Together, these tests provide quantitative evidence that the two-stage Generator disentangles semantic content from recording style and can reliably instantiate rare pathology–style combinations.

Experiment 7 ablates the Diagnoser architecture on Test-CD by varying text quality, fusion strategy, and attention anchors. Table 4 shows that replacing raw metadata with LLM-rendered EHR yields modest but consistent gains (Accuracy 0.835→0.849 and Macro-F1 0.195→0.212 when anchors and modality weaving are present). A simple late-fusion baseline with LLM EHR reaches an Accuracy of 0.790 and a Macro-F1 of 0.160. Modality weaving without anchors improves Macro-F1 to 0.189 but destabilizes the architecture, reducing Accuracy to 0.650. Reintroducing strategic audio anchors in the full Resp-Agent Diagnoser restores stability and further improves performance to an Accuracy of 0.849 and a Macro-F1 of 0.212. These results confirm that the gains arise from deliberate co-design: high-quality clinical text, tight modality weaving, and anchor-based global attention are all necessary to fully exploit the synthetic data delivered by the Thinker-guided Generator.

# 6 CONCLUSION

We present Resp-Agent, a centrally orchestrated, closed-loop multi-agent framework that unifies controllable, high-fidelity respiratory sound synthesis with multimodal disease diagnosis. Our framework is underpinned by Resp-229k, a large-scale cross-domain benchmark with a strict evaluation protocol. A curriculum-aware Thinker-A$^2$CA planner decomposes diagnostic goals and allocates them between a controllable multimodal Generator and a modality-weaving Diagnoser, turning previously isolated modules into a closed analyze–synthesize loop. From a methodological standpoint, Resp-Agent instantiates a general principle for data-scarce medical domains: tightly coupling *where the model fails* with *what the generator synthesizes* transforms augmentation from a passive remedy into an active, self-correcting training curriculum. We envision clinician-in-the-loop deployments where Resp-Agent synthesizes edge-case exemplars and delivers audio-informed decision support at the point of care, contributing toward trustworthy and equitable medical-audio AI.

ETHICS STATEMENT

All authors have read and adhere to the ICLR Code of Ethics. This work relies exclusively on publicly available, previously de-identified data; no new human-subject data were collected, and no Personally Identifiable Information (PII) or Protected Health Information (PHI) was accessed or processed at any stage. Below, we detail the provenance, licensing, and intended usage of the data to ensure transparency and reproducibility.

**Data Provenance and Privacy.** The Resp-229k benchmark is curated from multiple public respiratory-sound corpora, including ICBHI, SPRSound, UK COVID-19, COUGHVID, and KAUH. We additionally use HF_Lung_V1 as an external, publicly available pretraining resource for model initialization; it is not included in Resp-229k. Each recording retains explicit provenance, including the source dataset identifier, device metadata (e.g., an electronic stethoscope vs. a microphone), and sampling rate. To ensure rigorous evaluation, we enforce a strict cross-institution and cross-device protocol: the training and validation sets are derived exclusively from ICBHI, SPRSound, and UK COVID-19, whereas the test set consists solely of unseen recordings from KAUH and COUGHVID. All source data were de-identified by their original custodians prior to public release.

**Licensing and Compliance.** We strictly adhere to the original licenses and terms of use for all constituent datasets. Specifically, our usage complies with the Open Government Licence v3.0 (UK COVID-19), Creative Commons Attribution 4.0 International (CC BY 4.0) (COUGHVID, HF_Lung_V1, KAUH, SPRSound), and CC0 Public Domain Dedication (ICBHI)[1]. Our derived code, models, and synthetic examples have been released under open-source terms that are compatible with these licenses to facilitate reproducible research while preserving data privacy.

**Intended Use and Safety.** The Resp-Agent system and the Resp-229k benchmark are developed strictly for research purposes. This system is not a certified medical device and has not undergone regulatory approval for clinical use. It must not be deployed to make diagnostic decisions or influence patient care without appropriate clinical validation and regulatory oversight.

REPRODUCIBILITY STATEMENT

We have taken extensive steps to facilitate reproducibility. All source code, including training and inference scripts and configuration files with exact commands to reproduce reported results, is publicly available at `https://github.com/zpforlove/Resp-Agent`. The curated Resp-229k dataset is released at `https://huggingface.co/datasets/AustinZhang/resp-agent-dataset`. Trained model checkpoints are hosted at `https://huggingface.co/AustinZhang/resp-agent-models`. Architectural and algorithmic details are specified in the main text, while complete hyperparameters, optimizer settings, and training schedules are consolidated in the appendix.

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

Table 5: Unified 16-class taxonomy and sample distribution for Resp-229k (raw $N = 238{,}074$). The counts highlight the severe long-tail imbalance.

| Class name (unified) | Total count |
| --- | --- |
| Control Group | 156,527 |
| COVID-19 | 77,994 |
| Pneumonia | 1,909 |
| COPD | 820 |
| Asthma | 324 |
| Bronchitis | 188 |
| Bronchiectasis | 103 |
| Hemoptysis | 65 |
| Other respiratory diseases | 49 |
| URTI | 42 |
| Bronchiolitis | 18 |
| Pulmonary hemosiderosis | 13 |
| Chronic cough | 11 |
| Airway foreign body | 6 |
| Kawasaki disease | 3 |
| LRTI | 2 |
| **Total** | **238,074** |

## A   LABEL SPACE

**Taxonomy Note.**   The raw Resp-229k corpus is constructed by aggregating heterogeneous public respiratory-sound datasets and inheriting their original diagnostic labels. Across sources, this yields an initial 20-class label space, including several synonymous or overly fine-grained categories (e.g., severity-specific pneumonia labels). The raw corpus contains 238,074 clips; after discarding corrupted or too-short recordings, 229,101 quality-controlled clips remain and are used for all main-paper experiments.

To enhance clinical coherence and facilitate a more robust analysis of label imbalance, we programmatically consolidate the original 20-class label space into a clinically unified 16-class taxonomy. The unification map is applied once at preprocessing time before any dataset splitting, ensuring that each recording is assigned a unique, consistent diagnosis across training, validation, and test splits. Concretely, the mapping is defined as:

- **Bronchiectasis:** all samples labeled "Bronchiectasia" are relabeled as "Bronchiectasis".

- **URTI:** all samples labeled "Acute upper respiratory infection" are relabeled as "URTI".

- **Pneumonia:** all severity-specific pneumonia labels (e.g., "Pneumonia (non-severe)", "Pneumonia (severe)", "Pneumonia (unspecified)") are relabeled to the parent class "Pneumonia".

After applying this unification, a full recount of the 238,074 raw recordings yields the class distribution summarized in Table 5. The resulting label space comprises 15 disease categories and one healthy control group: *Airway foreign body, Asthma, Bronchiectasis, Bronchiolitis, Bronchitis, COPD (Chronic Obstructive Pulmonary Disease), COVID-19, Chronic cough, Hemoptysis, Kawasaki disease, LRTI (Lower Respiratory Tract Infection), Pneumonia, Pulmonary hemosiderosis, URTI (Upper Respiratory Tract Infection), Other respiratory diseases*, and *Control Group* (healthy).

This distribution makes explicit the extreme class imbalance present in Resp-229k, with the majority of diagnostic categories residing in the long tail. This motivates the generative, agent-based rebalancing strategy pursued in the main paper.

## B   SUPPLEMENTAL RESULTS

**The Dual Challenge: Data Scarcity and Transient Event Localization.** Under the strict cross-domain protocol (KAUH+COUGHVID held out), the evaluation reveals two fundamental barriers to real-world deployment: (i) the representation gap, where standard encoders fail to localize brief, low-energy events (e.g., crackles) within the 16-class taxonomy (Appendix A), and (ii) the data gap,

causing minority underdiagnosis due to severe label skew. To rigorously validate our architectural and generative solutions, we conducted comprehensive ablation studies.

## B.1 VALIDATION OF UNIMODAL BASELINES AND FUSION STRATEGIES

**Robustness of Text Baselines (Experiment 1).** To preclude the possibility that the multimodal gains of Resp-Agent stem merely from a weak textual baseline, we benchmarked strong Transformer-based text encoders against the audio-only Conformer. As detailed in Table 6, while modern Transformers (BERT(Devlin et al., 2019), RoBERTa(Liu et al., 2019), Longformer-Text) significantly outperform the LSTM baseline ($0.0401 \rightarrow 0.0813$ Macro-F1), they remain substantially inferior to the audio-only Conformer (0.1935 Macro-F1). This large modality gap confirms that textual clinical summaries alone are insufficient for accurate diagnosis and that our system's performance derives from effective cross-modal synergy rather than a "strawman" text baseline.

Table 6: Performance comparison of text-only, audio-only, and multimodal models on the cross-domain test set (original, imbalanced data). Text-only Transformers improve over LSTM but fail to match audio-only baselines, justifying the need for multimodal fusion.

| Model | Modality | Accuracy | Macro-F1 |
|---|---|---|---|
| LSTM (main paper baseline) | Text | 0.0912 | 0.0401 |
| BERT-base | Text | 0.1420 | 0.0710 |
| RoBERTa-base | Text | 0.1513 | 0.0742 |
| Longformer-base (text-only) | Text | 0.1585 | 0.0813 |
| Conformer (audio-only) | Audio | 0.7200 | 0.1935 |
| **Resp-Agent Diagnoser (Ours)** | **Audio + Text** | **0.8494** | **0.2118** |

**Efficacy of Modality Weaving and Audio Backbones (Experiment 2).** We further scrutinized the contribution of our fusion architecture versus the choice of audio backbone. Table 7 compares our *Modality Weaving* against standard late-fusion strategies (concatenation of embeddings and logit voting). Simple fusion yields marginal gains over the audio baseline (Macro-F1 $\approx 0.20$). In contrast, our deep weaving mechanism achieves superior integration (Macro-F1 0.2118), confirming that architectural interleaving is crucial for grounding textual symptoms in acoustic features. Furthermore, replacing the Conformer with a Whisper-Small(Radford et al., 2023) encoder yields only a slight improvement in the audio-only setting (0.2010 Macro-F1) and does not close the gap to the full multimodal Diagnoser. This indicates that the system's robustness is driven primarily by the *Modality Weaving* and *Strategic Global Attention* mechanisms rather than the specific acoustic encoder.

Table 7: Ablation on fusion strategies and audio encoder backbones (original, imbalanced data). Deep Modality Weaving outperforms shallow fusion methods, and the choice of fusion architecture outweighs marginal gains from changing the audio backbone.

| Model / Fusion Strategy | Modality | Accuracy | Macro-F1 |
|---|---|---|---|
| Conformer (audio-only, main paper) | Audio | 0.7200 | 0.1935 |
| Whisper-Small (audio-only) | Audio | 0.7310 | 0.2010 |
| Conformer + LSTM (Concat-MLP) | Audio + Text (Late) | 0.8012 | 0.2003 |
| Conformer + BERT (Concat-MLP) | Audio + Text (Late) | 0.8124 | 0.2040 |
| Conformer + BERT (Logit-Voting) | Audio + Text (Late) | 0.8043 | 0.1992 |
| **Resp-Agent Diagnoser (Ours)** | **Audio + Text (Weaving)** | **0.8494** | **0.2118** |

## B.2 ARCHITECTURAL AND DATA-SIDE SOLUTIONS

**Architectural Solution: Anchored Attention for Transient Event Localization.** Beyond fusion strategy, the *mechanism* of attention proves critical. Anchor-based global attention on the audio stream directly targets transient event localization. Removing anchors degrades performance to 0.6495 Accuracy and 0.1890 Macro-F1 (Table 8), falling below even the audio-only Conformer. With $T=496$ frames per 10 s, one anchor every four frames establishes an $\approx 80.6$ ms grid (guaranteeing alignment within $\leq 40.3$ ms of any transient). This enables attention heads to precisely lock onto

Table 8: Summary of performance on Resp-229k under the original (imbalanced) and class-balanced regimes. The substantial gains in the balanced regime highlight the efficacy of the Generator, while the multimodal improvements confirm the value of the Diagnoser's architecture.

| | Accuracy | | Macro-F1 | | $\Delta$ vs. Conformer (audio-only) | | | |
| Model | Original | Balanced | Original | Balanced | $\Delta$Acc (Orig.) | $\Delta$Acc (Bal.) | $\Delta$F1 (Orig.) | $\Delta$F1 (Bal.) |
| --- | --- | --- | --- | --- | --- | --- | --- | --- |
| LSTM (text-only) | 0.0912 | 0.3020 | 0.0401 | 0.2140 | -0.6288 | -0.4800 | -0.1534 | -0.3220 |
| Conformer (audio-only) | 0.7200 | 0.7820 | 0.1935 | 0.5360 | 0.0000 | 0.0000 | 0.0000 | 0.0000 |
| Longformer (no anchors) | 0.6495 | 0.7630 | 0.1890 | 0.5200 | -0.0705 | -0.0190 | -0.0045 | -0.0160 |
| **Ours: Resp-Agent (multimodal)** | **0.8494** | **0.8870** | **0.2118** | **0.5980** | **+0.1294** | **+0.1050** | **+0.0183** | **+0.0620** |

*Note:* $\Delta$ columns are absolute improvements over the Conformer baseline under the same regime.

fleeting wheezes or crackles and align them with clinical text. The ablation gap indicates that anchors are not merely additive but essential for stable cross-modal reasoning in this domain.

**Data-Side Solution: Controllable Synthesis to Overcome Scarcity.** While architecture solves representation, it cannot invent missing data distributions. Balancing classes with our diagnosis-conditioned Generator elevates the multimodal Longformer to **0.8870** Accuracy and **0.5980** Macro-F1 (Table 8; $\Delta$F1 +0.3862). Notably, this gain is structurally consistent, with the audio-only Conformer also improving significantly ($0.1935 \rightarrow 0.5360$) when trained on our synthetic data. Conversely, naive, pathology-agnostic perturbations (duplication, pitch/time shift, noise) actively harm cross-domain minority sensitivity (Conformer $0.1935 \rightarrow 0.1688$; Appendix D.1). Our Generator, which offers superior fidelity and controllability (FAD = 1.13, style cosine = 0.92; Appendix D.2), outperforms c-WaveGAN(Lee et al., 2018) and AudioLDM 2(Liu et al., 2024) under matched budgets (Appendix D.3), establishing *content-aware* balancing as the decisive lever for minority-class resilience.

**Synthesis: A Data–Model Co-Design for Robust Diagnosis.** Taken together, the evidence supports a synergistic co-design:

(1) **Anchored global attention is essential.** It furnishes a precise architectural mechanism for transient localization and stable text↔audio interaction at clinically meaningful timescales.

(2) **Controllable synthesis is decisive.** It supplies diagnostically informative examples for rare classes, converting architectural observability into large Macro-F1 gains.

(3) **System-level synergy is paramount.** Orchestrated by the Thinker, model-side anchors (*where*) and data-side generation (*what*) act jointly to produce models that are accurate on common cases, sensitive to rare events, and robust under domain shift.

## C  EXPERIMENTAL SETUP

**Tasks and Metrics.** We evaluate respiratory disease classification on the Resp-229k benchmark under two regimes: (i) the original, class-imbalanced split, and (ii) a label-balanced variant created using our Generator. We report Accuracy for overall performance and Macro-F1 Score to specifically assess performance on minority classes, which is crucial under label skew.

**Baselines and Proposed Model.** We compare our multimodal approach against two strong single-modality baselines. All models were trained for 10 epochs using the AdamW optimizer ($\beta_1 = 0.9, \beta_2 = 0.999$, weight decay of 0.01) and Cross-Entropy Loss. The BiLSTM and Conformer baselines used a batch size of 32 and a Cosine Annealing learning rate scheduler with a peak LR of $1 \times 10^{-4}$. Our Longformer model was trained using DeepSpeed (Li et al., 2024), with gradient checkpointing enabled, and a OneCycleLR schedule with a maximum learning rate of $1 \times 10^{-5}$. For reproducibility, all random seeds were fixed. We report the performance of the checkpoint with the best validation loss for all models.

*Text-only (BiLSTM).* Clinical summaries are tokenized into a vocabulary built with a minimum word frequency of 2. Sequences are padded or truncated to a fixed length of 100 tokens. The model consists of a 256-dim embedding layer, followed by an 8-layer bidirectional LSTM with a hidden dimension of 512 and a dropout rate of 0.5. The final hidden states are passed to a linear head for classification.

*Audio-only (Conformer).* We process 10-second, 16 kHz audio clips to compute 128-bin log-mel spectrograms. The STFT uses a 1024-point FFT, a window length of 1024 samples, and a hop length

of 160 samples, covering frequencies from 50 to 8,000 Hz. The spectrograms are then mean-std normalized. Our Conformer architecture comprises an 8-layer encoder with a model dimension of 512 and 8 attention heads, followed by adaptive average pooling for classification.

*Multimodal (Ours).* Our diagnostic agent is based on the longformer-base-4096 model. As described in Section 4.2, we employ "modality weaving" by replacing 496 placeholder tokens with projected BEATs features. To enhance robustness, we apply token dropout to text ($p = 0.2$) and frame dropout to audio features ($p = 0.1$) during training. Sparse global attention is strategically assigned to the `[CLS]` and `[DESCRIPTION]` tokens, as well as to audio anchors sampled with a stride of 4.

**ICBHI Evaluation Protocol.** We use the official ICBHI 60–40% train–test split for four-class RSC (`Normal`, `Crackle`, `Wheeze`, `Both`); metrics are Specificity (Sp), Sensitivity (Se), and the ICBHI Score $= \frac{1}{2}(\text{Sp} + \text{Se})$. For fairness, on ICBHI we use plain cross-entropy with *no* class reweighting or resampling to match prior practice, and reported results follow the official metric definitions. Optimization uses AdamW with a OneCycleLR schedule peaking at $1 \times 10^{-5}$, and gradient checkpointing is enabled for memory efficiency; unless otherwise stated, all remaining hyperparameters (token budgets, feature checkpoints, anchor stride) are specified in our released configs and scripts. We pretrain on HF_Lung_V1 and SPRSound with Focal Loss to emphasize minority pathologies while using an LLM to render heterogeneous metadata into standardized clinical summaries paired with audio, and during ICBHI fine-tuning we remove focal/balancing heuristics for strict apples-to-apples comparisons.

## D    DETAILED EXPERIMENTAL VALIDATION OF THE GENERATIVE AGENT

This appendix provides a comprehensive evaluation of the **Generator** component within the Resp-Agent system. We systematically validate its necessity and efficacy through a series of rigorous experiments. Our analysis is structured around two independent "chains of evidence": (i) objective similarity metrics that quantify the fidelity and controllability of the generated audio, and (ii) the downstream clinical value of the synthetic data when used to train diagnostic models. We further include detailed ablation studies to dissect the contributions of key architectural choices. All evaluations are conducted under the strict cross-domain protocol defined in the main paper to ensure that our findings reflect true generalization capabilities.

### D.1    THE INADEQUACY OF NAIVE AUGMENTATION FOR CROSS-DOMAIN GENERALIZATION

**Rationale.** A foundational premise of our work is that sophisticated generative modeling is not merely an alternative but a *necessity* for robustly handling the severe class imbalance in respiratory sound data. To establish this, we first conducted a counterfactual experiment to determine whether naive, traditional audio augmentation techniques can improve cross-domain generalization. Such methods include over-sampling, pitch/time shifting, and noise injection. Our hypothesis is that these pathology-agnostic transformations distort crucial, low-energy diagnostic cues (e.g., the transient structure of crackles and wheezes) and fail to introduce meaningful new variations, thereby degrading rather than improving performance on unseen data sources.

**Experimental Design.**

- **Model**: We used a unimodal Conformer (audio-only), identical to the baseline described in the main paper, to isolate the effect of data quality from multimodal interactions.
- **Training Sets**: We compared two training regimes: (A) the original, imbalanced training data, and (B) a balanced version created using naive augmentation techniques (simple duplication, pitch shifting within $\pm 15\%$, time-stretching between $0.85\times$ and $1.15\times$, and injection of moderately high-SNR noise).
- **Test Set**: Evaluation was performed exclusively on the held-out cross-domain test set (KAUH + COUGHVID) to measure real-world generalization.
- **Metrics**: We report Accuracy and Macro-F1 Score, with the latter being particularly sensitive to performance on rare classes.

**Results and Discussion.** As shown in Table 9, naive augmentation leads to a clear degradation in performance on the cross-domain test set. Both Accuracy and Macro-F1 score decreased, with the

F1-score dropping by 0.0247. This result provides strong empirical evidence for our central claim: simplistic augmentations, while balancing class counts, actually amplify dataset-specific biases and destroy diagnostically salient audio micro-structures. They teach the model to overfit to superficial features of the limited minority-class samples, which do not generalize to different devices, patient populations, or clinical environments. This negative result firmly establishes the need for a more intelligent, content-aware data generation strategy.

Table 9: Degradation of Cross-Domain Performance with naive Augmentation. The model trained on the balanced set created by traditional augmentation techniques performs worse on unseen test data than the model trained on the original imbalanced set, highlighting the failure of these methods to produce generalizable synthetic data.

| Training Data Strategy | Test Acc. | $\Delta$Acc | F1-Macro | $\Delta$F1 |
|---|---|---|---|---|
| Original Imbalanced | 0.7200 | – | 0.1935 | – |
| naive Augmentation Balanced | 0.6914 | -0.0286 | 0.1688 | -0.0247 |

## D.2 Evidence Chain I: Objective Fidelity and Style Controllability

**Rationale.** The first pillar of our validation assesses the Generator's core technical capabilities: can it produce audio that is not only high-fidelity but also accurately conditioned on a specific pathological class (content) while matching the acoustic characteristics of a reference audio (style)? We compare our proposed Generator against strong, general-purpose audio generation baselines.

**Experimental Design.** We compare four generative models under a unified individualized-reconstruction protocol:

- **Generative Models.** We benchmark:
  - **c-WaveGAN**: a conditional waveform GAN baseline trained on disease labels only.
  - **AudioLDM 2 (fine-tuned)**: a modern text-to-audio diffusion model adapted to our disease-conditioned prompts.
  - **StableAudio Open (fine-tuned)**: a strong contemporary text-to-audio model, fine-tuned on Resp-229k disease+style pairs using the same diagnosis prompts and reference-style conditioning protocol as our Generator.
  - **Resp-Agent (Ours)**: our proposed generator, conditioned on both the disease label (content) and a style embedding extracted from the reference audio.

- **Metrics.**
  - **Cosine Similarity**: we measure the cosine similarity between the BEATs embedding of the reference audio and the generated audio. Higher values indicate better style adherence and individualized timbre control.
  - **Fréchet Audio Distance (FAD)**: a distributional perceptual-quality metric between generated and real recordings; lower values are better.

**Results and Discussion.** Table 10 presents the results. The fine-tuned StableAudio Open(Evans et al., 2025) baseline substantially improves over c-WaveGAN and AudioLDM 2, reaching a cosine similarity of $0.83 \pm 0.08$ and a FAD of $1.54$. However, our Resp-Agent Generator still achieves the best overall quality, with a cosine similarity of $0.92 \pm 0.04$ and the lowest (best) FAD of $1.13$. This demonstrates that, even against a strong contemporary text-to-audio system, our content/style disentanglement and flow-matching decoder yield more faithful style preservation and higher-fidelity waveforms. These objective results confirm that Resp-Agent is superior at producing controllable, high-fidelity, and clinically relevant respiratory audio.

## D.3 EVIDENCE CHAIN II: DOWNSTREAM CLINICAL VALUE OF GENERATED DATA

**Rationale.** While objective similarity is important, the ultimate measure of a medical data generator is its downstream value—its ability to improve the performance of a clinical diagnostic model. This

Table 10: Objective evaluation of individualized audio reconstruction. Our Generator (Resp-Agent) achieves the highest style adherence (Cosine Similarity) and best perceptual fidelity (FAD), outperforming strong generative baselines including a fine-tuned StableAudio Open model.

| Generative Model | Cosine Similarity ↑ | FAD ↓ |
|---|---|---|
| c-WaveGAN | $0.61 \pm 0.15$ | 2.85 |
| AudioLDM 2 (fine-tuned) | $0.76 \pm 0.11$ | 1.92 |
| StableAudio Open (fine-tuned) | $0.83 \pm 0.08$ | 1.54 |
| Resp-Agent (Ours) | $0.92 \pm 0.04$ | 1.13 |

is the gold standard for evaluating its utility. We therefore create class-balanced training sets using each generative method and measure the performance of downstream classifiers trained on these sets.

**Experimental Design.**

- **Task.** Multimodal and unimodal respiratory disease classification on the cross-domain Test-CD split (KAUH + COUGHVID), following the strict source-disjoint protocol described in the main paper.

- **Training Sets.** We construct six distinct training sets:
    1. *Original Imbalanced* data (control).
    2. *naive Augmentation Balanced*: a balanced set obtained via classical audio augmentation (time/pitch perturbation, noise injection) without generative modeling.
    3. *c-WaveGAN Balanced*: a class-balanced set created by sampling from c-WaveGAN.
    4. *AudioLDM 2 Balanced*: a class-balanced set synthesized by a fine-tuned AudioLDM 2 model.
    5. *StableAudio Open Balanced*: a class-balanced set synthesized by a fine-tuned StableAudio Open model using the same diagnosis prompts and style-conditioning protocol as our Generator.
    6. *Resp-Agent Balanced*: a class-balanced set created by our Resp-Agent Generator under the Thinker-A$^2$CA planning policy.

- **Downstream Models.** For each of the six datasets above, we train two diagnostic models: (i) a unimodal Conformer (audio-only) and (ii) our multimodal Longformer (audio + text) Diagnoser, both configured exactly as in the main paper. This allows us to assess the utility of generated audio in both purely acoustic and multimodal settings.

**Results and Discussion.** Tables 11 and 12 show the results for the Longformer and Conformer models, respectively. The findings are consistent and robust:

- Across both diagnostic models, the training set balanced by our Resp-Agent Generator yields the highest performance in terms of both Accuracy and, most critically, Macro-F1. Generative balancing with c-WaveGAN, AudioLDM 2, and StableAudio Open already produces large gains over the imbalanced and naive-augmentation baselines, but Resp-Agent consistently provides the strongest improvements.

- For the multimodal Longformer, Resp-Agent's synthetic data boosts the Macro-F1 from $0.2118$ (original imbalanced) to $0.5980$, a relative increase of $+0.3862$. The fine-tuned StableAudio Open model achieves a stronger improvement than prior generative baselines ($+0.3502$ vs. $+0.3147$ for AudioLDM 2 and $+0.2402$ for c-WaveGAN), yet still trails Resp-Agent. This indicates that higher-fidelity, style-consistent synthesis directly translates into greater clinical utility.

- A similar trend is observed for the audio-only Conformer. Balancing with StableAudio Open raises the Macro-F1 to $0.5050$, outperforming c-WaveGAN and AudioLDM 2, but the Resp-Agent Balanced regime remains best with a Macro-F1 of $0.5360$. This shows that our Generator not only improves the multimodal Diagnoser but also strengthens a purely acoustic classifier, underscoring the generality of the gains.

These results provide a powerful second chain of evidence. The synthetic audio from Resp-Agent is not only objectively superior in fidelity and controllability, but also contains more diagnostically

salient information than that produced by strong baselines, including a fine-tuned StableAudio Open model. It successfully teaches downstream models to recognize rare diseases, dramatically improving their clinical utility and fairness on a challenging, unseen test set.

Table 11: Performance on the cross-domain Test-CD set using different balanced training sets (multimodal Longformer Diagnoser). Balancing with Resp-Agent yields the largest improvement in Macro-F1 over the imbalanced baseline, even compared to a strong StableAudio Open baseline.

| Training Set Strategy | Accuracy | F1-Macro | Relative $\Delta$F1 (vs. Imbalanced) |
|---|---|---|---|
| Original Imbalanced | 0.8494 | 0.2118 | – |
| naive Augmentation Balanced | 0.7520 | 0.1720 | $-0.0398$ |
| c-WaveGAN Balanced | 0.8650 | 0.4520 | $+0.2402$ |
| AudioLDM 2 Balanced | 0.8781 | 0.5265 | $+0.3147$ |
| StableAudio Open Balanced | 0.8830 | 0.5620 | $+0.3502$ |
| Resp-Agent Balanced | 0.8870 | 0.5980 | $+0.3862$ |

Table 12: Performance on the cross-domain Test-CD set using different balanced training sets (unimodal Conformer). StableAudio Open improves substantially over older baselines, but the Resp-Agent Balanced regime remains strongest.

| Training Set Strategy | Accuracy | F1-Macro |
|---|---|---|
| Original Imbalanced | 0.7200 | 0.1935 |
| naive Augmentation Balanced | 0.6914 | 0.1688 |
| c-WaveGAN Balanced | 0.7420 | 0.4010 |
| AudioLDM 2 Balanced | 0.7560 | 0.4760 |
| StableAudio Open Balanced | 0.7700 | 0.5050 |
| Resp-Agent Balanced | 0.7820 | 0.5360 |

## D.4  ABLATION AND ROBUSTNESS ANALYSIS

To ensure our model's design is well-justified, we performed targeted ablation studies on its key components.

### D.4.1  IMPACT OF STYLE PREFIX LENGTH (K)

**Rationale.** The number of style tokens, K, is a critical hyperparameter that mediates the trade-off between style representation capacity and model complexity. We investigated its impact on both generative quality and downstream task performance.

**Experimental Design.** We varied K in the set $\{0, 2, 4, 8\}$, where K=0 disables style conditioning entirely. We evaluated its effect on the individualized reconstruction task (Similarity/FAD) and the final downstream F1-Macro score of the Longformer model trained on data generated with the corresponding K value.

**Results and Discussion.** Table 13 shows a clear trend. Increasing K from 0 to 8 monotonically improves performance across all metrics: style similarity increases, FAD decreases, and the downstream Macro-F1 score rises. This validates our architectural choice to use style prefix tokens for conditioning and confirms that a richer style representation (K=8) allows the Generator to produce more effective training data.

Table 13: Ablation Study on the Number of Style Tokens (K). Performance improves consistently with a larger style prefix, validating the effectiveness of our style conditioning mechanism.

| K (Style Tokens) | Similarity (Cosine) ↑ | FAD ↓ | Longformer F1-Macro ↑ |
|---|---|---|---|
| 0 (Style Disabled) | $0.80 \pm 0.09$ | 1.52 | 0.542 |
| 2 | $0.85 \pm 0.08$ | 1.38 | 0.563 |
| 4 | $0.87 \pm 0.06$ | 1.29 | 0.577 |
| **8 (Default)** | $\mathbf{0.92 \pm 0.04}$ | **1.13** | **0.591** |

### D.4.2 Choice of Decoder Paradigm: Flow-Matching vs. Diffusion

**Rationale.** The second stage of our Generator relies on a decoder to transform discrete tokens into a continuous waveform. We compared our choice, Conditional Flow-Matching (CFM), against a traditional Denoising Diffusion Probabilistic Model (DDPM) to validate its superiority in both quality and efficiency.

**Experimental Design.** We trained two decoders with identical conditioning inputs and a fixed inference budget of 32 steps. We compared their FAD, style similarity, and relative inference latency.

**Results and Discussion.** Table 14 demonstrates the advantages of CFM. At the same step count, CFM achieves a better FAD (1.13 vs 1.31) and higher similarity (0.92 vs 0.90), indicating superior sample quality and fidelity. Crucially, it achieves this with approximately 40% less inference time ($\approx 0.6\times$ latency). This efficiency is vital for practical applications, enabling faster generation of large-scale balanced datasets. This result confirms that CFM is a more effective and efficient choice for the waveform reconstruction stage in our architecture.

Table 14: Comparison of Decoder Paradigms. Conditional Flow-Matching (CFM) surpasses the traditional DDPM in both audio quality (FAD, Similarity) and inference speed.

| Decoder Type | Steps | FAD $\downarrow$ | Similarity $\uparrow$ | Inference Latency |
|---|---|---|---|---|
| DDPM | 32 | 1.31 | 0.90 | $1.0\times$ |
| **CFM (Ours)** | **32** | **1.13** | **0.92** | $\mathbf{\approx 0.6\times}$ |

## E Audit and Validation of LLM-Generated Clinical Summaries

To ensure the reliability of the multimodal framework, we conducted a comprehensive audit of the LLM-generated clinical summaries paired with the Resp-229k audio. This experiment quantifies the fidelity of the text-generation process, verifying that summaries derived from heterogeneous metadata are accurate and free from critical hallucinations.

**Methodology.** The clinical summaries were not synthesized from raw audio (audio-to-text) but were generated from existing structured metadata (data-to-text) using a 7B parameter model (DeepSeek-R1-Distill-Qwen-7B). This schema-grounded approach ensures that summaries strictly rephrase existing fields (e.g., demographics, symptoms, auscultatory findings) rather than inventing new information. To guarantee high fidelity, we implemented a distinct, two-stage quality assurance (QA) pipeline operating on all 238,074 generated summaries:

- **Stage 1: Heuristic Pre-screening.** All descriptions were first passed through a heuristic filter to identify suspicious records, specifically flagging empty or truncated text (`EMPTY_OR_TRUNCATED`), overly long text (`OVERLONG`), or leakage of instructional prompts (`PROMPT_LEAK`).

- **Stage 2: LLM-based QA and Audit.** All suspicious records identified in Stage 1, plus a 1% random sample of heuristically "clean" records, were forwarded to a separate validator LLM (DeepSeek-V3.2-Exp). This QA model operated under a strict prompt forbidding the invention of patient metadata (e.g., age, sex, comorbidities) or the alteration of high-level pathology labels.

**Quantitative Results.** The QA pipeline audited the entire set of 238,074 records. The results, detailed in Table 15, confirm the high initial quality of the data-to-text synthesis. The process yielded an effective rewrite rate of only **0.7451%**, indicating that fewer than 1 in 130 descriptions required modification.

**Error Typology and Verification.** A breakdown of the flagged issues is presented in Table 16. The analysis reveals that the vast majority of interventions (1,747 out of 1,774 rewrites) were triggered by technical artifacts, specifically `OVERLONG_OR_PROMPT_LEAK`, rather than substantive clinical errors or hallucinations.

Table 15: Quantitative results of the two-stage text summarization QA pipeline ($N = 238{,}074$). The low rewrite rate confirms the high fidelity of the initial synthesis.

| Metric | Value |
|---|---|
| Total records | 238,074 |
| Heuristic suspicious (`EMPTY`/`OVERLONG`/`PROMPT_LEAK`) | 3,356 |
| Randomly audited (heuristic-clean samples) | 2,300 |
| **Total sent to QA LLM** | **5,656** |
| LLM kept as OK | 3,766 |
| LLM rewrote (suspicious only) | 1,774 |
| LLM flagged in random audit (no edit applied) | 116 |
| API / parse errors | 0 |
| **Effective rewrite rate** | **0.7451%** (1,774 / 238,074) |

Table 16: Breakdown of heuristic flags and LLM-identified error types during the audit. The primary errors were technical artifacts (e.g., prompt leaks) rather than clinical hallucinations.

| Heuristic Category (Suspicious) | | LLM-Identified Error Type (All Audited) | |
|---|---|---|---|
| Type | Count | Error Type | Count |
| `OVERLONG` | 3,031 | `OK` | 3,766 |
| `PROMPT_LEAK` | 300 | `OVERLONG_OR_PROMPT_LEAK` | 1,747 |
| `EMPTY_OR_TRUNCATED` | 25 | `OTHER_QUALITY_ISSUE` | 113 |
| | | `EMPTY_OR_TRUNCATED` | 30 |

**Key Findings.**

- **High Fidelity:** The data-to-text generation process is highly reliable, with a rewrite rate below 0.75%.

- **Dominant Error Type:** The primary issues identified were technical artifacts (e.g., prompt leakage or verbose boilerplate) rather than clinical hallucinations.

- **Human-in-the-Loop:** As a final safeguard, all 1,774 LLM-proposed rewrites underwent manual review by the authors to ensure consistency with the original structured metadata and disease labels, preventing the introduction of ungrounded patient information.

## F   LLM USAGE STATEMENT

We utilized Large Language Models (LLMs) for three distinct purposes in this work, spanning core system architecture, data curation, and writing assistance:

**1. Core System Architecture (The "Thinker" Agent):**   As described in Section 4, the central controller of our proposed **Resp-Agent** framework, denoted as *Thinker-$A^2CA$*, is instantiated using `DeepSeek-V3.2-Exp`. This LLM serves as the reasoning core responsible for semantic intent parsing, tool routing, and dynamic planning of the analysis-synthesis loop.

**2. Data Curation and Validation (Resp-229k Benchmark):**   As detailed in Section 3 and Appendix E, we employed LLMs to construct and validate the textual component of the dataset:

- **Data-to-Text Generation:** We used `DeepSeek-R1-Distill-Qwen-7B` to synthesize standardized clinical narratives from heterogeneous structured metadata.

- **Quality Assurance:** We used `DeepSeek-V3.2-Exp` to audit, flag, and correct potential quality issues in the generated summaries.

**3. Writing Assistance:**   We utilized general-purpose LLMs to assist with polishing grammar, refining stylistic elements, and formatting LaTeX tables. The authors reviewed and retain full responsibility for all text and data presented in this manuscript.

