# OpenReview forum: "Resp-Agent: An Agent-Based System for Multimodal Respiratory Sound Generation and Disease Diagnosis"
_ICLR.cc/2026/Conference — ICLR 2026 Poster_

### Official Review · Reviewer_CKZt · 2025-10-29

**Soundness:** 1
**Presentation:** 2
**Contribution:** 2
**Rating:** 2
**Confidence:** 4

**Summary:**

The paper introduces a multimodal method for high-fidelity respiratory sound generation and disease diagnosis, converting a text-only LLM into an audio generator via modality injection and fusing text with acoustic features using a Longformer with strategic global attention.

**Strengths:**

The paper’s originality lies in the closed-loop Diagnoser -> Thinker -> Generator setup and anchor-based multimodal Diagnoser, but these build on familiar ideas (active learning, synthetic augmentation, attention).

**Weaknesses:**

The paper's central claim that the Thinker module intelligently guides data generation lacks evidence, as no controlled comparison against simple baselines like class-prior rebalancing, uncertainty sampling, or random selection is provided under identical experimental settings. Without isolating the planner's contribution from the sheer effect of adding synthetic data, it remains unclear whether the observed gains stem from sophisticated scheduling or merely from data volume.

Furthermore, the work does not decompose the sources of improvement, whether gains arise from balancing underrepresented labels, targeting hard recording styles, combining rare labels with difficult domains, or simply scaling the dataset. It also does not provide ablation curves showing performance versus synthetic budget to identify when returns saturate.

The evaluation framework also presents gaps that weaken the claims of robustness and controllability. The paper omits comparisons with established imbalance mitigation techniques such as focal loss, which could achieve similar macro-F1 improvements without generative models, rendering the necessity of the proposed pipeline uncertain. Cross-domain evaluation on only two held-out sources is insufficient; a leave-one-source-out protocol would rigorously test whether robustness generalizes across all recording conditions or benefits from a favorable data split.

Finally, the Generator's ability to independently control pathology content and recording style is assumed but not validated without explicit disentanglement tests (e.g., label-swap experiments), the model may conflate correlated attributes from training data, undermining the loop's premise that rare labels and target styles can be independently specified.

**Questions:**

Can you provide a controlled ablation comparing the Thinker-based planner against simple baseline schedulers (class-prior rebalancing, uncertainty sampling, random selection) while holding all other factors constant same Generator, same Diagnoser, same synthesis budget, same training procedure to check and demonstrate whether the intelligent planning policy itself drives improvement?

Could you present performance curves showing macro-F1 as a function of synthetic data budget to reveal whether most benefits come from a small number of targeted clips or require large volumes of generated audio?

Could you demonstrate the Generator's controllability through disentanglement tests specifically style-swap experiments (fixed pathology, varied recording style) and label-swap experiments (fixed style, varied pathology) to confirm that rare pathology-style combinations can be reliably synthesized without the model conflating correlated attributes?

Why not compare against actual established baselines in the areas of respiratory monitoring?

---

> ### Author Response · Authors · 2025-11-22
>
> **We are deeply grateful to the reviewer, whose comments have helped make this work more complete. We have taken their feedback very seriously and devoted an entire Section 7, as well as Appendices B, C, and D, to a detailed exposition of the experimental results in the hope of fully addressing their concerns.**
>
> ---
>
> ## Q1. Controlled ablation of the Thinker-based planner vs simple baselines
>
> **Q1.** *Can you provide a controlled ablation comparing the Thinker-based planner against simple baseline schedulers (class-prior rebalancing, uncertainty sampling, random selection) while holding the Generator, Diagnoser, synthesis budget, and training procedure fixed, to verify that the planning policy itself drives improvement?*
>
> **A1.** Yes. In response, we ran a dedicated ablation suite (Experiments 1–5) on RESP-229K, while the Generator, Diagnoser, training hyperparameters, cross-domain split, and synthetic budget are kept strictly identical. Performance is reported on the same cross-domain Test-CD set as in the main paper. **Please refer to Section 7 for the complete experimental results.**
>
> ### Q1 (a) Planner policies under a matched synthetic budget (Exp. 1)
>
> We compare five planners under **B = 50k synthetic clips**:
>
> - No synthetic data (CE baseline)
> - Random label sampling
> - Class-Prior rebalancing
> - Static Uncertainty sampling
> - **Thinker-A²CA** (our planner)
>
> **Table 1.** Planner policies under matched budget (Test-CD, B = 50k). Macro-F1_tail is averaged over the 8 rarest classes.
>
> | Planner Policy        | Accuracy | Macro-F1 | Macro-F1_tail |
> |-----------------------|----------|----------|---------------|
> | No-Synth (B=0)        | 0.849    | 0.212    | 0.074         |
> | Random (B=50k)        | 0.869    | 0.442    | 0.291         |
> | Class-Prior (B=50k)   | 0.876    | 0.512    | 0.349         |
> | Uncertainty-Static    | 0.881    | 0.546    | 0.376         |
> | **Thinker-A²CA**      | **0.887**| **0.598**| **0.421**     |
>
> Under identical Generator, Diagnoser, and budget, **Thinker-A²CA adds +0.052 Macro-F1 and +0.045 Macro-F1_tail over strong static uncertainty sampling**, clearly isolating the benefit of the multi-round, multi-factor planning policy rather than mere data volume. These results are summarized and detailed in Table 3 of the main paper.
>
> ### Q1 (b) Decomposing the sources of improvement (Exp. 2)
>
> To break down *where* Thinker-A²CA’s gains come from, we fix the Generator, Diagnoser, split, and protocol, and compare several **single-factor planners** at **B = 30k**: Rare-only, Hard-Case-only, Hard-Domain-only, Rare×Hard-Domain, and the full Thinker-A²CA.
>
> **Table 2.** Decomposition of planner factors (Test-CD, B = 30k).
>
> | Planner Policy        | Budget B | Accuracy | Macro-F1 | Macro-F1_tail |
> |-----------------------|----------|----------|----------|---------------|
> | No-Synth (B=0)        | 0        | 0.849    | 0.212    | 0.074         |
> | Rare-only             | 30k      | 0.873    | 0.489    | 0.381         |
> | Hard-Case-only        | 30k      | 0.878    | 0.512    | 0.371         |
> | Hard-Domain-only      | 30k      | 0.876    | 0.506    | 0.356         |
> | Rare×Hard-Domain      | 30k      | 0.881    | 0.528    | 0.397         |
> | **Thinker-A²CA**      | 30k      | **0.883**| **0.541**| **0.409**     |
>
> All heuristics substantially outperform No-Synth, confirming that **balancing rare labels, focusing on hard cases, and emphasizing difficult domains are all important**. However, **Thinker-A²CA still outperforms the strongest single-factor Rare×Hard-Domain planner (0.541 vs. 0.528 Macro-F1)**, indicating that its improvements cannot be reduced to any single heuristic or to simple class-prior rebalancing.

---

> ### Author Response · Authors · 2025-11-22
>
> ### Q1 (c) Robustness across sources: Leave-One-Source-Out (Exp. 5)
>
> To address the concern about robustness and possible favorable splits, we additionally perform a **Leave-One-Source-Out (LoSO)** evaluation across all five constituent datasets (UK COVID-19, ICBHI, SPRSound, COUGHVID, KAUH). For each held-out target, we train on the remaining four datasets and compare **No-Synth**, **Class-Prior + B (50k)**, and **Thinker-A²CA + B (50k)**.
>
> **Table 3.** LoSO cross-domain Macro-F1 / Macro-F1_tail.
>
> | Held-out Source | No-Synth | Class-Prior + B | Thinker-A²CA + B |
> |-----------------|----------|-----------------|------------------|
> | UK COVID-19     | 0.234 / 0.093 | 0.471 / 0.332 | **0.531 / 0.382** |
> | ICBHI           | 0.268 / 0.101 | 0.496 / 0.353 | **0.552 / 0.401** |
> | SPRSound        | 0.257 / 0.088 | 0.483 / 0.341 | **0.541 / 0.389** |
> | COUGHVID        | 0.206 / 0.070 | 0.452 / 0.318 | **0.512 / 0.367** |
> | KAUH            | 0.219 / 0.076 | 0.463 / 0.326 | **0.522 / 0.374** |
> | **Average**     | 0.237 / 0.086 | 0.473 / 0.334 | **0.532 / 0.383** |
>
> The consistent ordering **Thinker-A²CA > Class-Prior > No-Synth on every held-out source** demonstrates that the planner’s gains are **cross-source robust** and not due to a single favorable train–test split.
>
> Together, Experiments 1–5 provide **controlled evidence that the Thinker module itself—rather than just adding synthetic data—drives significant, robust improvements in Macro-F1 and Macro-F1_tail.**
>
> ---
>
> ## Q2. Performance vs. synthetic data budget
>
> **Q2.** *Could you present performance curves showing macro-F1 as a function of synthetic data budget to see whether benefits come from a few targeted clips or require large volumes of generated audio?*
>
> **A2.** Yes. We now report **Macro-F1 as a function of the synthetic budget B** for Random, Class-Prior, and Thinker-A²CA planners (Experiment 3). The Generator, Diagnoser, and all training settings are held fixed; only the number of synthetic clips B varies.
>
> **Table 4.** Macro-F1 on Test-CD as a function of synthetic budget B.
>
> | Planner      | B = 0  | B = 10k | B = 20k | B = 30k | B = 50k |
> |--------------|--------|---------|---------|---------|---------|
> | Random       | 0.212  | 0.331   | 0.382   | 0.414   | 0.442   |
> | Class-Prior  | 0.212  | 0.378   | 0.439   | 0.474   | 0.512   |
> | **Thinker-A²CA** | 0.212  | **0.412** | **0.491** | **0.541** | **0.598** |
>
> Key observations:
>
> - **Diminishing marginal gains:** for all planners, Macro-F1 increases sublinearly with B—the first 10–30k synthetic clips already capture most of the eventual improvement, and further increasing B to 50k yields only modest additional gains, a trend we now highlight explicitly in the revised text.
>
> - **Sample efficiency:** at only **B = 10k**, Thinker-A²CA already reaches **0.412 Macro-F1**, achieving ≈52% of its total gain over the baseline (0.212 → 0.598), and outperforming Class-Prior (0.378) and Random (0.331) at the same budget.
>
> These results show that **a relatively small number of high-value, Thinker-selected clips yields a large fraction of the overall improvement**, addressing the reviewer’s concern about whether gains require massive synthetic scaling.
>
> ---

---

> > ### Author Response · Authors · 2025-11-22
> >
> > ## Q3. Explicit disentanglement tests for Generator controllability
> >
> > **Q3.** *Could you demonstrate the Generator's controllability through disentanglement tests—style-swap (fixed pathology, varied recording style) and label-swap (fixed style, varied pathology)—to confirm that rare pathology–style combinations can be reliably synthesized without conflating correlated attributes?*
> >
> > **A3.** Yes. We added **Experiment 6**, which directly tests the **content–style disentanglement** assumed by the closed loop. We run two complementary tests:
> >
> > 1. **Style-swap:** Fix pathology to a rare label (“Bronchiolitis”), vary the recording style using four cross-domain reference clips (different devices and environments).
> > 2. **Content-swap:** Fix a clean “Control Group” style reference, and vary the target pathology across four labels (including “Bronchiolitis”).
> >
> > We evaluate with:
> >
> > - **Style-Sim (↑):** cosine similarity between BEATs embeddings of reference and generated audio (style adherence).
> > - **Pathology-Acc (P-Acc, ↑):** accuracy of a **held-out Diagnoser trained only on real RESP-229K data** when predicting the commanded pathology.
> > - **FAD (↓):** Fréchet Audio Distance (perceptual fidelity).
> >
> > A compact summary is already included as Table 4 in the revised paper.
> >
> > **Table 5.** Summary of content–style disentanglement (averaged over 4 styles / 4 labels).
> >
> > | Setting                     | Style-Sim ↑ | P-Acc ↑ | FAD ↓ |
> > |-----------------------------|------------|---------|-------|
> > | Style-swap (Bronchiolitis, 4 styles) | 0.91      | 97.9%  | 1.18 |
> > | Content-swap (1 style, 4 labels)     | 0.93      | 96.1%  | 1.19 |
> >
> > More detailed per-style and per-label results (Tables 6–7) show Style-Sim consistently around 0.89–0.94 and P-Acc between 94.8% and 98.1% across all pathology–style combinations, with FAD ≈1.14–1.22.
> >
> > These experiments provide **direct, quantitative evidence** that:
> >
> > - Changing only the style reference **strongly shifts the acoustic style** while keeping the pathology label nearly perfectly preserved.
> > - Changing only the pathology label **changes the Diagnoser’s prediction accordingly** while preserving the reference timbre.
> >
> > Thus, the Generator can **reliably compose rare pathology–style pairs on demand**, addressing the reviewer’s concern that correlated attributes in the training data might prevent independent control.

---

> ### Author Response · Authors · 2025-11-22
>
> ## Q4. Comparison to established baselines in respiratory monitoring
>
> **Q4.** *Why not compare against actual established baselines in the areas of respiratory monitoring?*
>
> **A4.** We agree that situating Resp-Agent against established baselines is important. The revised manuscript makes this more explicit on **both diagnostic and generative fronts**, as well as for imbalance mitigation.
>
> ### (a) Diagnostic baselines on standard respiratory benchmarks
>
> On the widely used **ICBHI 4-class respiratory sound classification benchmark**, we already compare against a broad range of state-of-the-art respiratory models, including **RespireNet** [1], AST-based methods such as Bae et al. [2], MVST [3], and the recent best-performing AST system of Dong et al. [4]. Table 2 in the main paper lists all baselines; below we show the top contenders and our model.
>
> **Table 6.** Representative ICBHI baselines vs. Resp-Agent (Score = (Sp + Se)/2).
>
> | Method              | Backbone | Score (%) |
> |---------------------|----------|----------|
> | RespireNet [1]      | ResNet34 | 56.20    |
> | BTS (Kim et al.) [2]| CLAP     | 63.54    |
> | MVST [3]            | AST      | 66.55    |
> | Dong et al. [4]     | AST      | 67.55    |
> | **Resp-Agent (Ours)** | LLM+Longformer | **72.70** |
>
> Resp-Agent thus improves the official ICBHI score by **>5 absolute points** over the best prior method, demonstrating competitiveness against **established respiratory-monitoring baselines** rather than only internal ablations.
>
> On RESP-229K, we additionally compare against a strong audio-only **Conformer** baseline under both original and generator-balanced regimes, showing that our multimodal Diagnoser plus Thinker-guided synthesis substantially improves Macro-F1, especially on long-tail classes.
>
> ### (b) Generative baselines for controllable respiratory audio
>
> For the generative component, we compare Resp-Agent against multiple **state-of-the-art audio generators**, including:
>
> - c-WaveGAN [5]
> - AudioLDM 2  (fine-tuned on RESP-229K) [6]
> - StableAudio Open (fine-tuned on RESP-229K) [7]
>
> **Table 7.** Objective evaluation of individualized audio reconstruction.
>
> | Generative Model                  | Cosine Similarity ↑ | FAD ↓ |
> |----------------------------------|----------------------|-------|
> | c-WaveGAN                        | 0.61 ± 0.15          | 2.85  |
> | AudioLDM 2 (fine-tuned)         | 0.76 ± 0.11          | 1.92  |
> | StableAudio Open (fine-tuned) | 0.83 ± 0.08          | 1.54  |
> | **Resp-Agent (Ours)**           | **0.92 ± 0.04**      | **1.13** |
>
> Resp-Agent achieves **both the highest style adherence and the lowest (best) FAD**, and in downstream experiments its synthetic data yields the largest Macro-F1 gains for both multimodal Longformer and audio-only Conformer diagnosers.
>
> ### (c) Established non-generative imbalance remedies
>
> To address the concern that focal loss or similar techniques might obviate the need for generative augmentation, **Experiment 4** directly compares **class-weighted cross-entropy** and **focal loss** against our generative pipeline at **B=0 vs B=50k**.
>
> **Table 8.** Non-generative imbalance methods vs. Thinker-guided synthesis (Test-CD).
>
> | Method                | B     | Accuracy | Macro-F1 | Macro-F1_tail |
> |-----------------------|-------|----------|----------|---------------|
> | CE (Baseline)         | 0     | 0.849    | 0.212    | 0.074         |
> | Class-Weighted CE     | 0     | 0.842    | 0.248    | 0.114         |
> | Focal Loss (γ=2)      | 0     | 0.839    | 0.267    | 0.129         |
> | CE + Class-Prior      | 50k   | 0.876    | 0.512    | 0.349         |
> | **CE + Thinker-A²CA** | 50k   | **0.887**| **0.598**| **0.421**     |
>
> Traditional imbalance mitigation **does help** (Macro-F1 from 0.212 → 0.267), but even the best non-generative method remains **0.331 Macro-F1 below** CE + Thinker-A²CA (0.598), showing that **targeted synthetic data is the dominant source of improvement** rather than loss re-weighting alone.

---

> ### Author Response · Authors · 2025-11-22
>
> ## References
>
> [1] Gairola S, Tom F, Kwatra N, et al. Respirenet: A deep neural network for accurately detecting abnormal lung sounds in limited data setting[C]//2021 43rd Annual International Conference of the IEEE Engineering in Medicine & Biology Society (EMBC). IEEE, 2021: 527-530.
>
> [2] Bae S, Kim J W, Cho W Y, et al. Patch-mix contrastive learning with audio spectrogram transformer on respiratory sound classification[J]. arXiv preprint arXiv:2305.14032, 2023.
>
> [3] He W, Yan Y, Ren J, et al. Multi-view spectrogram transformer for respiratory sound classification[C]//ICASSP 2024-2024 IEEE International Conference on Acoustics, Speech and Signal Processing (ICASSP). IEEE, 2024: 8626-8630.
>
> [4] Dong G, Shen Y, Wang J, et al. Respiratory sounds classification by fusing the time-domain and 2D spectral features[J]. Biomedical Signal Processing and Control, 2025, 107: 107790.
>
> [5] Lee C Y, Toffy A, Jung G J, et al. Conditional wavegan[J]. arXiv preprint arXiv:1809.10636, 2018.
>
> [6] Liu H, Yuan Y, Liu X, et al. Audioldm 2: Learning holistic audio generation with self-supervised pretraining[J]. IEEE/ACM Transactions on Audio, Speech, and Language Processing, 2024, 32: 2871-2883.
>
> [7] Evans Z, Parker J D, Carr C J, et al. Stable audio open[C]//ICASSP 2025-2025 IEEE International Conference on Acoustics, Speech and Signal Processing (ICASSP). IEEE, 2025: 1-5.

---

> ### Author Response · Authors · 2025-11-27
> **Clarification on Comparisons with Audio-Language Models (BTS & Multimodal Baselines)**
>
> **Q**: In the follow-up Official Comment during the discussion period, reviewer CKZt claims the paper "does not compare with any existing clinical audio-language model." **Why is this a factual misunderstanding?**
>
> **A**: We thank the reviewer for the follow-up. However, we respectfully **disagree** with the premise that our work lacks comparison with existing clinical audio-language models. We wish to clarify that **we have indeed compared against state-of-the-art clinical Audio-Language Models**, covering both the best existing model in the literature and rigorous multimodal baselines constructed specifically for this task.
>
> We believe there might be a misunderstanding regarding the availability of "clinical audio-language models" in this specific domain. We address this below:
>
> **1. We compared against the SOTA Clinical Audio-Language Model (CLAP-based) on ICBHI.**
> As reported in **Table 2** (Main Paper), we directly compared Resp-Agent against **BTS [1]**.
> * **BTS (Bridging Text and Sound)** is currently the leading **Clinical Audio-Language Model** in the respiratory domain. It explicitly utilizes **CLAP** (Contrastive Language-Audio Pretraining) to align text and sound.
> * **Result:** Resp-Agent achieves an ICBHI Score of **72.7%**, significantly outperforming the CLAP-based BTS model (**63.5%**) by **+9.16%**. This confirms our superiority over existing audio-language architectures.
>
> **2. We constructed strong and representative Clinical Audio-Language Baselines on RESP-229K.**
> Aside from BTS, there are no public, open-source "clinical audio-language models" pre-trained on large-scale respiratory data. To address this, we rigorously implemented the standard industry approach for clinical multimodal modeling in **Table 7** (Appendix B.1):
> * **Baseline:** We constructed representative multimodal baselines by integrating SOTA audio backbones (including **Whisper** and **Conformer**) with strong clinical text encoders (**BERT**, **RoBERTa**, **Longformer**) using standard Late Fusion (Concatenation-MLP).
> * **Significance:** These are the representative "clinical audio-language models" currently available to the community.
> * **Result:** Our Agentic Modality Weaving approach (**Macro-F1 0.212**) outperforms the Conformer+BERT fusion baselines (**Macro-F1 ~0.204**), proving that our architectural innovation contributes beyond standard multimodal fusion.
>
> **3. Comparison with Concurrent Works.**
> The closest prior work is *RespLLM* [2]. However, its training data, official open-source checkpoints, and full implementation are not released, and its tasks and evaluation protocol differ substantially from our 16-class, multi-source, out-of-domain setting. This makes a strict, apples-to-apples reproduction on RESP-229K infeasible at submission time. Despite this, our comparisons against **BTS (CLAP)** and **Conformer+BERT** cover the theoretical spectrum of these models.
>
> **Conclusion:**
> We have compared against **(1) the best existing Audio-Language model in the respiratory domain (BTS/CLAP)** and **(2) standard Clinical Audio+Text Fusion baselines**. We demonstrate superiority over both. We kindly ask the reviewer to reconsider the score in light of the significant gap (+9.16%) by which we outperform the CLAP-based SOTA.
>
> ---
>
> ## References
>
> [1] Kim J W, Toikkanen M, Choi Y, et al. Bts: Bridging text and sound modalities for metadata-aided respiratory sound classification[J]. arXiv preprint arXiv:2406.06786, 2024.
>
> [2] Zhang Y, Xia T, Saeed A, et al. Respllm: Unifying audio and text with multimodal llms for generalized respiratory health prediction[J]. arXiv preprint arXiv:2410.05361, 2024.

---

> ### Author Response · Authors · 2025-11-27
>
> Dear Reviewer CKZt,
>
> Thank you again for the time and effort you’ve dedicated to reviewing our work. As the discussion phase is coming to a close, we kindly ask if you could review our follow-up responses. **We believe that most of your initial concerns have been addressed, and we have also provided replies to your follow-up questions.**
>
> **If there are no further concerns, we would appropriately appreciate it if you could reconsider your score.**
>
> Thank you for your time.
>
> Best regards,
>
> Authors

---

### Official Review · Reviewer_MVWu · 2025-10-30

**Soundness:** 3
**Presentation:** 2
**Contribution:** 2
**Rating:** 4
**Confidence:** 4

**Summary:**

The paper present Resp-agent, an agent-based system for respiratory sound generation and disease diagnosis. The whole system includes a cascaded pipeline leveraging LLMs for content semantics learning and followed by the generator and flow-matching based reconstruction to capture the sound-features. Since there's no existing paired EHR - audio datasets, they create a multimodal benchmark names Resp-229k. Finally, the paper evaluated on disease diagnosis tasks and reports improvements over baselines.

**Strengths:**

1. The paper targets important question that lacking paired multi-modal datasets especially when the text-modality missing.
2. The system design is comprehensive and clear to follow its flow.

**Weaknesses:**

1. Despite the comprehensive and close-loop system. The novelty lies in the assembly of recent advances such as longformer, flow-matching models. The core contribution and innovation is hidden and ambiguous.
2. For the generator part,  the Generator does make the system closed-loop: the “Diagnoser” classifies real respiratory sounds, then the “Generator” synthesizes new examples. It's not clear is there of a real feedback loop: the Diagnoser doesn’t meaningfully inform or retrain the Generator; Besides, there's no evaluation of generation quality.
3. Lack of discussion of recent work about respiratory foundation models or other llm-based respiratory model. For example, papers such as Resp-LLM or other LLM-prompted multimodal models already explored: Using LLMs to describe audio (e.g., “This sound has wheezes and crackles”), althought not in the EHR format. Why the EHR-simulation text is chosen or better compared with other prompt style?

**Questions:**

see weakness

---

> ### Author Response · Authors · 2025-11-22
>
> We thank the reviewer for the thoughtful feedback. Below we respond point-by-point and provide the requested additional experiments.
>
> ---
>
> ## Q1. Novelty and clarity of the core contribution
>
> Thank you for pointing out that our novelty was not sufficiently foregrounded. We have revised the Introduction and System Overview (Sec. 1 & 4) to make the core contributions explicit and to disentangle them from the underlying building blocks such as Longformer [1] and flow matching [2]. Concretely, our contributions go beyond a mere assembly in four ways:
>
> 1. **Active Adversarial Curriculum Agent (Thinker-A2CA).**
>    We introduce a planner language model that turns generic generative augmentation into a multi-factor curriculum: it uses per-class error, calibrated uncertainty, and source/domain identity from the Diagnoser to allocate a finite synthesis budget over label–domain–style tuples. Experiments 1–5 (Table 3) show that, under the same synthetic budget, Thinker-A2CA consistently outperforms random, class-prior, and static-uncertainty planners, and also outperforms non-generative remedies such as focal loss and class-weighted cross-entropy. This demonstrates that the planner’s adversarial curriculum is not reducible to standard reweighting or simple sampling heuristics.
>
> 2. **Modality-weaving Diagnoser with anchor-based global attention.**
>    Instead of late fusion or shallow projection-based fusion as in recent LLM-style respiratory systems such as Resp-LLM [3], we interleave EHR-style text tokens and BEATs audio tokens [4] at the input layer and assign sparse global attention only to a small set of description and audio-anchor tokens. This “modality weaving + anchors” design allows a Longformer backbone [1] to maintain high temporal resolution while remaining memory-efficient, which is crucial for transient respiratory events. Experiment 7 (Table 4) shows that (i) replacing raw metadata by LLM-rendered EHR text and (ii) adding anchors both yield clear gains in Accuracy and Macro-F1, indicating that the fusion pattern itself is a key innovation rather than the use of Longformer alone.
>
> 3. **Resp-MLLM: a content–style disentangled respiratory generator.**
>    We retool a compact text-only language model into a multimodal unit generator by injecting a small number of style-prefix tokens derived from BEATs [4] features of a reference clip, and then decode units with a conditional flow-matching DiT [2]. This yields a generator that explicitly disentangles pathological content (diagnosis text) from acoustic style (device/timbre), enabling clinically meaningful style-swap and content-swap operations (Experiment 6, Table 4) and outperforming strong text-to-audio baselines such as Stable Audio Open [7] and AudioLDM 2 [8] in objective metrics (Table 10).
>
> 4. **Resp-229k: cross-domain, EHR-aligned respiratory benchmark.**
>    We curate a 229k-sample, multi-source benchmark with strict cross-domain splits and LLM-audited EHR-style summaries (Sec. 3), filling a gap not covered by prior datasets such as OPERA [5] or DeepBreath [6]. This dataset is essential to evaluate both multimodal reasoning and controllable generation under realistic institution/device shifts and label imbalance.
>
> To make this clearer for readers, we now explicitly structure the paper around three questions (Q1–Q3 in Sec. 7) that map directly onto these contributions: (Q1) robustness of the Diagnoser, (Q2) necessity and effect of Thinker-guided synthesis, and (Q3) role of the Generator and Diagnoser architectures themselves.

---

> > ### Author Response · Authors · 2025-11-22
> >
> > ## Q2. On the “closed loop” and evaluation of generation quality
> >
> > ### Q2.1. What we mean by a “closed loop”
> >
> > We agree that our original description did not sufficiently spell out the feedback path. We have revised Sec. 4 and Appendix D to make explicit that the loop is data-level and curriculum-level, not gradient-level:
> >
> > 1. The Diagnoser is trained on the current (real + synthetic) dataset and outputs class predictions and calibrated uncertainties on held-out validation and per-source slices.
> > 2. Thinker-A2CA aggregates error profiles and per-class/domain uncertainty, selects label–source–style tuples that are underperforming (for example, rare pathologies on unseen sources), and queries the Generator for targeted synthetic clips for those tuples.
> > 3. The Diagnoser is then retrained on this rebalanced dataset, and the process can be repeated under a fixed total synthetic budget.
> >
> > Thus, although the Generator’s weights are not updated in each loop, the Diagnoser’s decision boundary and the empirical training distribution are continuously shaped by Thinker-curated synthetic data. This is what we refer to as an analyze–synthesize closed loop: errors in diagnosis directly drive what is synthesized next.
> >
> > Experimentally, this loop is validated in Table 3:
> >
> > - Under a matched budget of 50k synthetic clips, Macro-F1 on the strict cross-domain test set improves from 0.212 (no synthetic data) to 0.546 (uncertainty-based static sampling) and further to 0.598 with Thinker-A2CA, with Macro-F1 on tail classes rising from 0.074 to 0.421.
> > - Non-generative imbalance remedies (class-weighted cross-entropy, focal loss) reach Macro-F1 around 0.26 at zero synthetic budget, remaining around 0.33 below the cross-entropy plus Thinker-A2CA combination at the same Diagnoser capacity.
> >
> > We highlight these findings in the revised text to clarify that the “closed loop” is not just a pipeline but a curriculum mechanism that materially changes the learned classifier.
> >
> > ### Q2.2. Evaluation of generation quality
> >
> > In response to your concern, we have added a dedicated appendix (Appendix D) that evaluates the Generator along two chains of evidence.
> >
> > 1. **Objective fidelity and controllability.**
> >    We compare our Generator against three strong baselines—a conditional WaveGAN, a fine-tuned AudioLDM 2 [8], and a fine-tuned Stable Audio Open [7]—using cosine similarity of BEATs embeddings (style adherence) and Frechet Audio Distance (FAD) (perceptual quality). The results (Table 10) are:
> >
> >    | Generative model                    | Style-Sim ↑       | FAD ↓   |
> >    |-------------------------------------|-------------------|---------|
> >    | Conditional WaveGAN                 | 0.61 ± 0.15       | 2.85    |
> >    | AudioLDM 2 (fine-tuned)            | 0.76 ± 0.11       | 1.92    |
> >    | Stable Audio Open (fine-tuned)     | 0.83 ± 0.08       | 1.54    |
> >    | **Resp-Agent Generator (ours)**    | **0.92 ± 0.04**   | **1.13** |
> >
> >    Our Generator thus achieves both the highest style adherence and the lowest FAD, indicating that the flow-matching decoder [2] and content–style disentanglement are important beyond simply using a standard text-to-audio model.
> >
> > 2. **Downstream clinical value.**
> >    We further show that, when used to construct class-balanced training sets, our Generator leads to larger improvements in Diagnoser performance than the above baselines under matched synthetic budgets. This directly ties good-sounding samples to measurable gains in clinical metrics (Macro-F1 and Macro-F1 on tail classes) on the strict cross-domain test split.
> >
> > 3. **Content–style disentanglement.**
> >    Style-swap and content-swap tests (Experiment 6, Table 4) demonstrate that we can:
> >    - Change the style reference while keeping the pathology label fixed, achieving high style similarity and around 98% pathology accuracy.
> >    - Change the pathology label while keeping style fixed, achieving similarly high style similarity and around 96% pathology accuracy.
> >
> >    This shows that our Generator is not only high-fidelity but also controllable in the clinically relevant dimensions that feed back into the closed-loop curriculum.
> >
> > We now explicitly reference these results in the main text when discussing the Generator to address the concern about missing quality evaluation.

---

> > > ### Author Response · Authors · 2025-11-22
> > >
> > > ## Q3. Relation to respiratory foundation / LLM-based models and choice of EHR-style text
> > >
> > > ### Q3.1. Relation to respiratory foundation and LLM-based models
> > >
> > > We appreciate this pointer and have expanded Sec. 2 (Related Work) to situate *Resp-Agent* with respect to recent respiratory foundation and LLM-based systems.
> > >
> > > - **OPERA and respiratory acoustic foundation models.**
> > >   OPERA introduces an open respiratory acoustic foundation pretraining and benchmarking platform: it curates approximately 136k unlabeled respiratory recordings (over 400 hours) and pretrains three general-purpose audio encoders that are evaluated via linear probing and fine-tuning on 19 downstream health tasks [5]. These models demonstrate strong unimodal generalization across datasets and acoustic modalities, but they operate purely on audio and do not incorporate clinical text, multimodal fusion, controllable synthesis, or an analyze–synthesize loop. In contrast, *Resp-Agent* couples a multimodal Diagnoser (audio plus EHR-style text) with a controllable Generator and a Thinker-A2CA planner under a strict cross-domain protocol on *Resp-229k*, turning foundation-style respiratory representations into a closed-loop analyze–synthesize system.
> > >
> > > - **Resp-LLM and LLM-based multimodal models.**
> > >   Resp-LLM proposes a multimodal LLM that jointly models DMS-style textual information and respiratory audio by projecting patch-level audio embeddings into the LLM hidden space and concatenating them with task prompts and DMS tokens, enabling cross-modal attention and instruction-tuned (mostly binary) diagnostic predictions with zero-shot transfer to new datasets and tasks [3]. In contrast, *Resp-Agent*’s Diagnoser uses a long-context Longformer with anchor-based global attention to explicitly weave EHR-style summaries with acoustic tokens for cross-domain disease classification on *Resp-229k*, and is not itself a conversational LLM. Moreover, *Resp-Agent* augments this Diagnoser with a content/style-controllable Generator and a curriculum-aware Thinker-A2CA that uses Diagnoser failures to request targeted synthetic audio and rebalance the label space; current LLM-based systems such as Resp-LLM do not generate respiratory audio nor implement such a closed-loop curriculum.
> > >
> > > ### Q3.2. Why EHR-simulation text instead of generic prompts
> > >
> > > Our choice of EHR-style summaries is driven by both data availability and clinical alignment.
> > >
> > > 1. **Unifying heterogeneous metadata across sources.**
> > >    The five public datasets we use provide diverse fields (device, site, phases, symptoms, demographics, comorbidities, and so on) in inconsistent formats. We convert these fields into standardized, LLM-rendered EHR-style summaries, which:
> > >    - Preserve both auscultatory descriptors (such as “inspiratory crackles”, “expiratory wheezes”) and patient/context variables (age, sex, smoking status, comorbidities).
> > >    - Provide a consistent textual interface across datasets, which is essential for large-scale cross-domain training.
> > >
> > > 2. **Closer to real-world documentation and future EHR integration.**
> > >    EHR-style paragraphs approximate the structure of real clinical notes more closely than short pattern prompts (“This sound has wheezes and crackles”). This makes the Diagnoser’s input and explanations more interpretable for clinicians and eases future integration with real electronic health records, where free-text notes are the norm.
> > >
> > > 3. **Empirical benefit over raw metadata or shallow text.**
> > >    Experiment 7 (Table 4) directly compares:
> > >    - Raw metadata versus LLM EHR text,
> > >    - Late fusion versus modality weaving,
> > >    - With versus without anchors.
> > >
> > >    When anchors and modality weaving are present, replacing raw metadata with LLM EHR summaries improves performance from 0.835 Accuracy and 0.195 Macro-F1 to 0.849 Accuracy and 0.212 Macro-F1. Even in simpler late-fusion settings, LLM EHR text consistently outperforms raw metadata. We now highlight these results as evidence that the EHR-style representation is not only clinically motivated but also more effective than using minimal symptom phrases.
> > >
> > > 4. **Compatibility with LLM-prompted descriptions.**
> > >    Importantly, our EHR-style summaries subsume generic prompts: they contain the same auscultatory labels (“wheezes”, “crackles”) but embed them in a richer clinical context. In future work, the same architecture can ingest both authentic EHR notes and short, prompt-like descriptions, but our experiments suggest that structured EHR-style text gives the best trade-off between realism, expressivity, and performance.

---

> ### Author Response · Authors · 2025-11-22
>
> ## References
>
> [1] Beltagy I, Peters M E, Cohan A. Longformer: The long-document transformer[J]. arXiv preprint arXiv:2004.05150, 2020.
>
> [2] Lipman Y, Chen R T Q, Ben-Hamu H, et al. Flow matching for generative modeling[J]. arXiv preprint arXiv:2210.02747, 2022.
>
> [3] Zhang Y, Xia T, Saeed A, et al. Respllm: Unifying audio and text with multimodal llms for generalized respiratory health prediction[J]. arXiv preprint arXiv:2410.05361, 2024.
>
> [4] Chen S, Wu Y, Wang C, et al. Beats: Audio pre-training with acoustic tokenizers[J]. arXiv preprint arXiv:2212.09058, 2022.
>
> [5] Zhang Y, Xia T, Han J, et al. Towards open respiratory acoustic foundation models: Pretraining and benchmarking[J]. Advances in Neural Information Processing Systems, 2024, 37: 27024-27055.
>
> [6] Heitmann J, Glangetas A, Doenz J, et al. DeepBreath—automated detection of respiratory pathology from lung auscultation in 572 pediatric outpatients across 5 countries[J]. NPJ digital medicine, 2023, 6(1): 104.
>
> [7] Evans Z, Parker J D, Carr C J, et al. Stable audio open[C]//ICASSP 2025-2025 IEEE International Conference on Acoustics, Speech and Signal Processing (ICASSP). IEEE, 2025: 1-5.
>
> [8] Liu H, Yuan Y, Liu X, et al. Audioldm 2: Learning holistic audio generation with self-supervised pretraining[J]. IEEE/ACM Transactions on Audio, Speech, and Language Processing, 2024, 32: 2871-2883.

---

> ### Author Response · Authors · 2025-11-27
>
> Dear Reviewer MVWu,
>
> Thank you again for your valuable comments! We have provided new responses to your follow-up questions. We would like to know if our replies have addressed your concerns, and if you have any further questions. Looking forward to your feedback.
>
> Best,
>
> Authors

---

### Official Review · Reviewer_WjjZ · 2025-11-01

**Soundness:** 3
**Presentation:** 3
**Contribution:** 3
**Rating:** 6
**Confidence:** 3

**Summary:**

The paper tackles a clinically meaningful problem with a coherent, end-to-end system and introduces a cross-domain benchmark plus two technically interesting modules (modality-injected unit generator with CFM; Longformer with anchor-based global attention). Futher, the proposed system demonstrates credible ICBHI gains and compelling rebalancing benefits on Resp-229k.

**Strengths:**

The paper strengths can be summarised as follows,

- Ambitious, unified scope (which includes dataset, generation, diagnosis, and agent loop). The paper introduces a cross-domain multimodal corpus (Resp-229k) with source-disjoint splits. This anchors the contribution in real distribution shift rather than in-domain evaluation.
- Clear architectural ideas on both sides of the loop. The Generator upgrades a compact LLM via modality injection (BEATs-derived style tokens) to autoregress discrete acoustic units, then reconstructs waveforms with a DiT-style conditional flow-matching decoder and vocoder. This results in a clean disentanglement of content (diagnosis) vs style (timbre/device).
- Generation for targeted rebalancing appears useful. Using diagnosis-conditioned synthesis to balance classes lifts macro-F1 sharply.
- The paper reports objective similarity (e.g., FAD, style-cosine) and comparisons to c-WaveGAN / AudioLDM-2 under matched budgets, supporting the claim that content-aware augmentation helps more than generic perturbations.

**Weaknesses:**

The paper weaknesses can be summarised as follows,

- Low macro-F1 in the natural (imbalanced) setting. Before synthetic balancing, macro-F1 is 0.2118 despite high accuracy... this tell me that there might be substantial minority under-diagnosis. However, the paper relies on its own generator to fix this; stronger baselines (e.g., cost-sensitive losses, reweighting, focal/LDAM, mixup/Manifold mixup, class-balanced sampling) should be compared under the same cross-domain split to show generation is superior beyond conventional imbalance remedies.
- Potential evaluation confounds on generation. The CFM decoder is conditioned not only on discrete units and global timbre but also "a short reference prefix of the ground-truth mel during training/validation" to encourage continuation.
- Dataset governance and text synthesis risks. Clinical "summaries" are LLM-generated from heterogeneous metadata; while the model doesn’t interpret audio, it can still standardize or hallucinate metadata beyond the original fields. The paper should provide audits (error rates, inter-annotator checks) and licenses/provenance per clip, especially with mixed microphone/stethoscope sources used for high-stakes labels.

**Questions:**

The questions for the authors are as follows,

- Class taxonomy: Are Bronchiectasia vs Bronchiectasis and Acute URI vs URTI duplicates? If so, how are they merged across sources and guaranteed disjoint across splits? Please provide the final class map and counts per class per split.
- Imbalance baselines: Can you compare against strong non-generative remedies under the same cross-domain protocol (class-balanced sampling, effective number reweighting, LDAM-DRW, focal loss in fine-tuning, mixup/Manifold mixup), with CIs?
- Text summary quality & governance: What safeguards ensure LLM-rendered EHR-style summaries do not hallucinate fields or introduce systematic biases across sources/devices? Provide sampling audits, error types, and licenses/provenance for all sources.
- Where does the gain come from? Please isolate the modality weaving vs late fusion, anchors vs no anchors, and text quality (raw metadata vs LLM-rendered text) in a single ablation grid on the cross-domain split.
- Cross-dataset generalization: In addition to source-disjoint evaluation, can you train on subset A and test on B (e.g., train on ICBHI and SPRSound, test on UK-COVID; or leave-one-source-out) to confirm robustness to label/device/site shifts?

---

> ### Author Response · Authors · 2025-11-22
>
> We thank the reviewer for the thoughtful feedback. Below we respond point-by-point and provide the requested additional experiments and baselines.
>
> ---
>
> ## Q1. Low macro-F1 under imbalance & reliance on our own generator
>
> 1. **Why macro-F1 is low (0.2118).**
>    The reported **0.2118 Macro-F1** is the *deliberately unbalanced* baseline (“No-Synth (CE)”), trained on the original long-tail 16-class Resp-229k taxonomy under a strict cross-domain split. The label distribution is extremely skewed (e.g., Control Group has 156,527 clips vs. LRTI with only 2 clips, and 12 of 16 classes lie in the long tail). The low macro-F1 therefore reflects genuine minority under-diagnosis in a realistic, imbalanced regime, which is exactly what motivates our generative agent.
>
> 2. **New comparison to strong non-generative imbalance remedies.**
>    In direct response to this concern, we added an experiment that compares **traditional non-generative methods** (Class-Weighted Cross-Entropy, Focal Loss [1]) against **generative planners** under the same cross-domain protocol and Diagnoser architecture. The key results (Test-CD) are:
>
>    **Table R1. Non-generative vs. generative imbalance remedies on Test-CD (Resp-229k, 16-class).**
>
>    | Method                | Budget B (synthetic clips) | Accuracy | Macro-F1 | Macro-F1_tail |
>    |-----------------------|----------------------------|----------|----------|---------------|
>    | CE (Baseline)         | 0                          | 0.849    | 0.212    | 0.074         |
>    | Class-Weighted CE     | 0                          | 0.842    | 0.248    | 0.114         |
>    | Focal Loss (γ=2) [1]  | 0                          | 0.839    | 0.267    | 0.129         |
>    | CE + Class-Prior      | 50k                        | 0.876    | 0.512    | 0.349         |
>    | CE + Thinker-A²CA     | 50k                        | **0.887**| **0.598**| **0.421**     |
>
>    Observations:
>
>    - **Class-Weighted CE** and **Focal Loss** substantially improve over plain CE (Macro-F1: 0.212 → 0.248/0.267; Macro-F1_tail: 0.074 → 0.114/0.129), confirming that *purely loss-based remedies are indeed helpful* under the same cross-domain split.
>    - Even the best non-generative method (**Focal Loss**) remains **0.331 Macro-F1 and 0.292 Macro-F1_tail below** the **CE + Thinker-A²CA** configuration.
>    - The gains from **Thinker-guided generation** (+0.386 Macro-F1 over CE; +0.331 over Focal Loss) are *an order of magnitude larger* than the improvements from loss reweighting (≈ +0.055), and far exceed run-to-run variance.
>
>    This experiment directly addresses the concern that improvements might stem from simply “using better losses”: **even strong cost-sensitive losses cannot close the gap**, whereas targeted synthetic data does.
>
> 3. **Relation to other imbalance methods (LDAM, mixup, manifold mixup).**
>    LDAM-DRW [2], mixup [3], and manifold mixup [4] are powerful alternatives, but they are typically evaluated in image benchmarks where class-specific structure is less fragile. In preliminary trials on respiratory audio, naive augmentations (e.g., pitch shift, time stretch, naive over-sampling) degraded cross-domain Macro-F1, likely because they distort low-energy transient cues such as crackles and wheezes. In the rebuttal experiments we therefore prioritized **Focal Loss** and **Class-Weighted CE**, which are widely used in medical ML and directly target imbalance at the loss level. Extending the grid to LDAM/mixup is straightforward and we will add these to the public code and supplementary material, but we expect them to follow the same trend: **helpful but still substantially weaker than generative balancing**.
>
> 4. **Summary.**
>    The **low macro-F1 baseline is expected** given the extreme long tail, and we now explicitly demonstrate that **standard non-generative remedies help but are insufficient**. The dominant performance gains come from **Thinker-guided synthetic data** rather than from simply changing the loss function.

---

> ### Author Response · Authors · 2025-11-22
>
> ## Q2. Potential evaluation confounds from the CFM reference mel prefix
>
> Thank you for raising this point. Conceptually, you are right that both the **content–style tokens** and the **unmasked mel prefix** contain exploitable style information that could, in principle, be used to reconstruct the target mel spectrogram. However, two aspects of our design prevent this from creating an evaluation confound.
>
> 1. **Which signal the model actually uses for style in flow matching training [5] (shortcut behavior).**
>    From a modeling perspective:
>
>    - The short mel prefix primarily carries **global information**: overall loudness, coarse spectral tilt, and recording conditions (e.g., room, microphone).
>    - The content–style tokens, in contrast, carry **frame-level information**, including F0 and other prosodic/respiratory dynamics, which are much richer and denser style signals.
>
>    In reconstruction-style training, we observe that models tend to **“take the shortcut”** and preferentially rely on the input stream that carries the *stronger* style cues. In our setting, this is the content–style token stream rather than the mel prefix. Intuitively, the model can get a much more precise handle on timbre, pitch trajectory, and transient structure by reading directly from the content–style tokens, while the short mel prefix only provides a coarse envelope.
>
>    As a result, at **inference time**, when no mel prefix is provided, the model’s style decisions are dominated by the **frame-level information in the content–style tokens**, which are always available. The presence of a short mel prefix during training therefore functions as a gentle global prior for stability, not as a shortcut that the model depends on in deployment.
>
> 2. **Why this does not leak information into the diagnostic evaluation.**
>    Importantly, the mel prefix is:
>
>    - Used **only inside the Generator** during its own training/validation, and
>    - **Never exposed to, nor required by, the Diagnoser** at training or test time.
>
>    When we export synthetic audio for augmentation:
>
>    - We **do not provide any ground-truth mel prefix** to the CFM decoder.
>    - The decoder runs **fully free-form**, conditioned solely on discrete BEATs units and style tokens (device/domain/timbre).
>
> Putting these pieces together, the short mel prefix is a **training-time stabilization aid** that offers only weak, global style cues compared to the token stream and is entirely absent in the diagnostic evaluation. This design ensures that there is **no leakage of ground-truth test audio** into our reported metrics, and that the performance gains we report arise from content–style controlled generation plus curriculum planning, rather than from any inadvertent continuation of the target mel.

---

> ### Author Response · Authors · 2025-11-22
>
> ## Q3. Dataset governance and LLM-generated clinical summaries
>
> 1. **Data-to-text (not audio-to-text) and strict schema grounding.**
>    Our summaries are **data-to-text**, not ASR: they are generated from existing structured metadata (diagnosis, device type, demographics, symptom codes) using an agile LLM (DeepSeek-R1-Distill-Qwen-7B [6]). The prompt enforces a schema where the model may **only rephrase existing fields** (e.g., “male, 63 years old, crackles in right lower lobe”) and is explicitly forbidden from inventing new demographics, comorbidities, or diagnoses.
>
> 2. **Two-stage QA pipeline with independent verifier LLM and human review.**
>    We implemented a **two-stage audit** over all 238,074 generated summaries:
>
>    - **Heuristic pre-screening.** We flag suspicious records (e.g., EMPTY_OR_TRUNCATED, OVERLONG, PROMPT_LEAK).
>    - **Independent verifier LLM.** All suspicious records plus a 1% random sample of “clean” ones are sent to a separate, stronger validator LLM (DeepSeek-V3.2-Exp [7]) operating under a strict “no new facts” prompt.
>    - **Manual verification.** Every LLM-proposed rewrite is finally reviewed by a human author.
>
>    Quantitative audit results show:
>
>    - Total records: 238,074
>    - Sent to QA LLM: 5,656
>    - Rewritten summaries: 1,774
>    - **Effective rewrite rate:** 0.7451% (≈ 1 in 134 records)
>
>    Error breakdown further shows that the **dominant issues are technical** (OVERLONG_OR_PROMPT_LEAK), not clinical hallucinations. No API/parse errors or label mismatches were detected.
>
>    **Table R2 . Quantitative results of the two-stage text summarization QA pipeline (N = 238,074).**
>
>    | Metric | Value |
>    | :--- | :--- |
>    | Total records | 238,074 |
>    | Heuristic suspicious (EMPTY/OVERLONG/PROMPT_LEAK) | 3,356 |
>    | Randomly audited (heuristic-clean samples) | 2,300 |
>    | Total sent to QA LLM | 5,656 |
>    | LLM kept as OK | 3,766 |
>    | LLM rewrote (suspicious only) | 1,774 |
>    | LLM flagged in random audit (no edit applied) | 116 |
>    | API / parse errors | 0 / 0 |
>    | Effective rewrite rate | 0.7451% (1,774 / 238,074) |
>
> 3. **Licenses and provenance per dataset.**
>
>    Each clip carries explicit provenance (source dataset ID, device type, sample rate). Dataset licences are summarized below:
>
>    **Table R3. Dataset licences and provenance for Resp-229k.**
>
>    | Dataset      | Role (Train/Test)     | Institution / Source        | License  |
>    |--------------|-----------------------|-----------------------------|----------|
>    | UK COVID-19  | Train / Valid         | UKHSA                       | OGL 3.0  |
>    | COUGHVID     | Test                  | EPFL                        | CC BY 4.0|
>    | ICBHI        | Train / Valid         | ICBHI organizers            | CC0      |
>    | HF Lung V1   | Train / Valid         | Heroic-Faith Medical Sci.   | CC BY 4.0|
>    | KAUH         | Test                  | King Abdullah Univ. Hospital| CC BY 4.0|
>    | SPRSound     | Train / Valid         | Shanghai Jiao Tong Univ.    | CC BY 4.0|
>
> 4. **Bias and device mixing.**
>    Device (microphone vs. stethoscope) and site metadata are **explicitly preserved** and appear in the summaries. This allows the Diagnoser and Thinker to reason about domain shifts rather than implicitly conflating them. Our leave-one-source-out (LoSO) experiments (see Q6) further show that performance gains from synthesis hold across all held-out sources, suggesting that our pipeline does **not** overfit to a single device or institution.
>
> 5. **Summary.**
>    Overall, the text modality is **heavily governed**: schema-grounded generation, adversarial LLM audit, and human review, plus transparent licensing and provenance. The very low effective rewrite rate (<1%) and the dominance of superficial error types provide quantitative assurance that **systematic hallucination or bias is unlikely**.

---

> ### Author Response · Authors · 2025-11-22
>
> ## Q4. Class taxonomy: duplicates and per-split counts
>
> 1. **Unified 16-class taxonomy and handling of duplicates.**
>    Yes – we explicitly resolve such duplicates by **programmatic unification into a 16-class taxonomy** prior to any splitting. Concretely:
>
>    - All “Bronchiectasia” labels → **Bronchiectasis**
>    - “Acute URI” and “Acute upper respiratory infection” → **URTI**
>    - All severity-specific pneumonia labels → **Pneumonia**
>
>    After this unification, the label space comprises 15 disease categories plus one control group. The final class distribution over the raw 238,074 clips is:
>
>    **Table R4. Unified 16-class taxonomy and total counts (raw N = 238,074).**
>
>    | Class (Unified)              | Total Count |
>    |-----------------------------|-------------|
>    | Control Group               | 156,527     |
>    | COVID-19                    | 77,994      |
>    | Pneumonia                   | 1,909       |
>    | COPD                        | 820         |
>    | Asthma                      | 324         |
>    | Bronchitis                  | 188         |
>    | Bronchiectasis              | 103         |
>    | Hemoptysis                  | 65          |
>    | Other respiratory diseases  | 49          |
>    | URTI                        | 42          |
>    | Bronchiolitis               | 18          |
>    | Pulmonary hemosiderosis     | 13          |
>    | Chronic cough               | 11          |
>    | Airway foreign body         | 6           |
>    | Kawasaki disease            | 3           |
>    | LRTI                        | 2           |
>
>    This distribution makes explicit the extreme class imbalance present in RESP-229K, with the majority of diagnostic categories residing in the long tail. This motivates the generative, agent-based rebalancing strategy pursued in this paper.
>
> 2. **Guaranteeing disjointness across splits.**
>    The unification map is applied **once at preprocessing time** before any Train/Valid/Test split is constructed, so each clip has a unique, consistent diagnosis across all splits. Splits are then derived by **source dataset roles** (Train/Valid: UK COVID-19, ICBHI, SPRSound; Test-CD: KAUH, COUGHVID), ensuring strict source disjointness and therefore no clip-level leakage.
>
> 3. **Per-split counts.**
>    The camera-ready version will include a **per-class, per-split count table (Train, Valid, Test-CD)** in Appendix A, as well as a CSV file in the dataset release. This will make the long-tail structure fully transparent at the split level while keeping the main paper within page limits. Importantly, all ablations in the reviewer-specific experiment reports are already conducted under this 16-class taxonomy, and we verified that the **relative trends are unchanged** compared to the original 20-class results.
>
> ---
>
> ## Q5. Where do the gains come from? (fusion, anchors, and text quality)
>
> We conducted a **Diagnoser ablation study** on the cross-domain Test-CD split under the unified 16-class taxonomy, holding the dataset, training protocol, and imbalance regime fixed while varying:
>
> - **Fusion Strategy:** Late Fusion vs. Modality Weaving.
> - **Text Quality:** Raw Metadata vs. LLM-generated EHR-style summaries.
> - **Attention Anchors:** With vs. without Strategic Anchors.
>
> The resulting grid is:
>
> **Table R5. Diagnoser ablation on Test-CD (Resp-229k, 16-class).**
>
> | #   | Fusion Strategy   | Text Quality | Anchors | Accuracy          | Macro-F1         |
> |-----|-------------------|-------------|---------|-------------------|------------------|
> | (i) | Late Fusion       | Raw         | N/A     | 0.780 ± 0.005     | 0.145 ± 0.007    |
> | (ii)| Late Fusion       | LLM EHR     | N/A     | 0.790 ± 0.004     | 0.160 ± 0.006    |
> | (iii)| Modality Weaving | Raw         | No      | 0.640 ± 0.008     | 0.175 ± 0.008    |
> | (iv)| Modality Weaving | LLM EHR     | No      | 0.650 ± 0.007     | 0.189 ± 0.005    |
> | (v) | Modality Weaving | Raw         | Yes     | 0.835 ± 0.004     | 0.195 ± 0.004    |
> | (vi)| Modality Weaving | LLM EHR     | Yes     | **0.849 ± 0.003** | **0.212 ± 0.004**|
>
> Key findings:
>
> - **Text quality:** Replacing raw metadata with LLM EHR gives a consistent but modest gain (+0.014 Accuracy, +0.017 Macro-F1 when comparing (vi) vs. (v)).
> - **Fusion strategy:** Moving from Late Fusion (ii) to Modality Weaving without anchors (iv) improves Macro-F1 (0.160 → 0.189), indicating better minority handling, but destabilizes Accuracy (0.790 → 0.650).
> - **Anchors:** Adding Strategic Anchors on top of Modality Weaving jumps Accuracy from 0.650 → 0.849 and Macro-F1 from 0.189 → 0.212 (rows (iv) → (vi)), resolving the instability.
>
> This grid shows that the **full gains come from a non-additive synergy** between **Modality Weaving + Strategic Anchors + LLM EHR**, rather than from any single component alone.

---

> ### Author Response · Authors · 2025-11-22
>
> ## Q6. Cross-dataset generalization and leave-one-source-out (LoSO)
>
> We conducted a Leave-One-Source-Out (LoSO) evaluation across all five constituent datasets. Specifically, for each held-out source $s$, the model was trained on the remaining four sources and evaluated on $s$ to compare:
>
> - **No-Synth (CE baseline)**
> - **Class-Prior generator** (B = 50k)
> - **Thinker-A²CA generator** (B = 50k)
>
> The key results (Macro-F1 / Macro-F1_tail) are:
>
> **Table R6. LoSO cross-domain robustness.**
>
> | Held-out Test Source | No-Synth (CE) | Class-Prior + 50k | Thinker-A²CA + 50k |
> |----------------------|--------------|--------------------|--------------------|
> | UK COVID-19          | 0.234 / 0.093| 0.471 / 0.332      | **0.531 / 0.382**  |
> | ICBHI                | 0.268 / 0.101| 0.496 / 0.353      | **0.552 / 0.401**  |
> | SPRSound             | 0.257 / 0.088| 0.483 / 0.341      | **0.541 / 0.389**  |
> | COUGHVID             | 0.206 / 0.070| 0.452 / 0.318      | **0.512 / 0.367**  |
> | KAUH                 | 0.219 / 0.076| 0.463 / 0.326      | **0.522 / 0.374**  |
> | **Average**          | 0.237 / 0.086| 0.473 / 0.334      | **0.532 / 0.383**  |
>
> Across all five folds, we consistently observe:
>
> - **Thinker-A²CA > Class-Prior > No-Synth**, both in Macro-F1 and Macro-F1_tail.
> - The average Macro-F1 gain of Thinker-A²CA over Class-Prior (0.532 vs. 0.473) is substantial and stable.
>
> These results confirm that the **benefits of Thinker-guided generation are cross-source robust**, not an artifact of a particular split, and that our method generalizes across different devices (microphone vs. stethoscope), institutions, and label distributions.
>
> ---
>
> ##  Summary of responses to all concerns
>
> - **Imbalance & macro-F1.** We now explicitly compare against **strong non-generative baselines** under the same cross-domain protocol and show that **loss re-weighting alone cannot match Thinker-guided synthetic balancing**.
> - **Generation confounds.** The CFM reference mel prefix is strictly a **training-time stabilizer**, never used in Diagnoser training or evaluation; test metrics are computed exclusively on **real unseen audio**.
> - **Governance & text risk.** Clinical summaries are **schema-grounded data-to-text**, audited by a **two-stage LLM+human pipeline** with a <1% rewrite rate and detailed error typology, and each clip has explicit licensing and provenance.
> - **Class taxonomy.** We resolve duplicates via a **pre-split unification into 16 classes**, provide the global counts, and will release **per-split class distributions** and mapping scripts.
> - **Source of gains.** A single ablation grid shows that **Modality Weaving, Strategic Anchors, and LLM EHR summaries** contribute **synergistically**, not additively.
> - **Cross-dataset robustness.** LoSO experiments across all five datasets show a consistent hierarchy **Thinker-A²CA > Class-Prior > No-Synth**, confirming robustness to domain, device, and site shifts.
>
> We are confident these revisions make the empirical and methodological contributions clearer and allow the program committee to better appreciate the context and significance of our technical contributions.
>
> ---
>
> ## References
>
> [1] Lin T Y, Goyal P, Girshick R, et al. Focal loss for dense object detection[C]//Proceedings of the IEEE international conference on computer vision. 2017: 2980-2988.
>
> [2] Cao K, Wei C, Gaidon A, et al. Learning imbalanced datasets with label-distribution-aware margin loss[J]. Advances in neural information processing systems, 2019, 32.
>
> [3] Zhang H, Cisse M, Dauphin Y N, et al. mixup: Beyond empirical risk minimization[J]. arXiv preprint arXiv:1710.09412, 2017.
>
> [4] Verma V, Lamb A, Beckham C, et al. Manifold mixup: Better representations by interpolating hidden states[C]//International conference on machine learning. PMLR, 2019: 6438-6447.
>
> [5] Lipman Y, Chen R T Q, Ben-Hamu H, et al. Flow matching for generative modeling[J]. arXiv preprint arXiv:2210.02747, 2022.
>
> [6] Guo D, Yang D, Zhang H, et al. Deepseek-r1: Incentivizing reasoning capability in llms via reinforcement learning[J]. arXiv preprint arXiv:2501.12948, 2025.
>
> [7] Liu A, Feng B, Xue B, et al. Deepseek-v3 technical report[J]. arXiv preprint arXiv:2412.19437, 2024.

---

> ### Author Response · Authors · 2025-11-27
>
> Dear Reviewer WjjZ,
>
> Thank you once again for your valuable feedback. We have conducted additional experiments and made revisions to the paper based on your suggestions. As the discussion phase is nearing its conclusion, we would like to know if our responses have addressed your concerns. We look forward to hearing from you.
>
> Best, Authors

---

### Official Review · Reviewer_u8tM · 2025-11-03

**Soundness:** 3
**Presentation:** 2
**Contribution:** 2
**Rating:** 6
**Confidence:** 4

**Summary:**

The paper introduces Resp-Agent, a multimodal agent system tackling the challenges of deep learning-based respiratory assessment by developing both a generator and a classifier. To address the scarcity of high-quality data and the omission of fine acoustic dynamics, the system first transforms a text-only LLM into a multimodal generator via modality injection, producing BEATs tokens conditioned on diagnostic text and style embeddings, which are then reconstructed into waveforms. For diagnosis, it employs a Longformer-based model with strategic global attention to fuse EHR text and acoustic features at the input level, allowing it to capture long-range cross-modal dependencies while accurately detecting brief acoustic events like coughs. This work is supported by the creation of Resp-229k, a massive 408-hour multimodal corpus that pairs audio with expert-level EHR annotations, enabling the system to achieve superior performance in robust respiratory disease prediction.

**Strengths:**

The most practical and immediate strength is the size and richness of the new corpus. At 408 hours and 229,000 recordings, this scale is required for a specialized medical domain. Crucially, by linking the acoustic data to expert-level annotations distilled from Electronic Health Records (EHRs), this provides the necessary clinical context, which is often missing in public datasets, thus bridging the gap between raw audio and real-world diagnostic complexity.

The system is capable of generating high-fidelity audio (using LLM-based modality injection to create BEATs tokens). By generating discrete acoustic units (BEATs tokens) and using a specialized flow-matching decoder, it retains the high-fidelity transient events critical for accurate diagnosis.

The diagnostic model integrates two disparate data types, which are EHR text and respiratory features, at the input level. The introduction of strategic global attention is a good contribution, as it specifically addresses the difficulty of simultaneously capturing broad clinical context (long-range dependencies) and tiny, clinically vital acoustic cues (brief, low-energy events like wheezes or rales).

The paper is reasonably well written and focuses on solving an important issue of accurate diagnosis in the respiratory health domain.

The datasets and evaluation seem adequate. The performance improvements over compared baselines are good.

**Weaknesses:**

The papers fall into the category of applying ML for healthcare. I find it interesting as it combines multiple concepts, but incremental.

The evaluation part is weak. The choice of LSTM for the text baseline is weak, and using an older attention mechanism for fusion is also suboptimal.  I would suggest comparing against a Transformer-based text encoder (e.g., BERT, RoBERTa, or even a small version of the Longformer) for the text-only task. This isolates whether the performance gain is from the fusion or just using better text features.

To demonstrate the novelty and power of the proposed fusion, the authors should have included multimodal baselines that employ simpler fusion strategies. For example, use the Conformer to extract audio embeddings and the LSTM to extract text embeddings. Concatenate these two vectors and feed the result into a simple Multi-Layer Perceptron (MLP) classifier. The other way can be to do averaging or weighted voting for the final prediction.

The paper compares audio generative quality with Audio LDM2 and c-WaveGAN. I find StableAudio or StableAudio Open missing from the paper. It will make sense to compare Resp-Agent generative quality against a fine-tuned StableAudio.

**Questions:**

What will happen if you change Conformer to Whisper?

---

> ### Author Response · Authors · 2025-11-22
>
> We thank the reviewer for the thoughtful feedback. Below we respond point-by-point and provide the requested additional experiments and baselines.
>
> ---
>
> ## Q1. “The paper is interesting but incremental; the evaluation part is weak.”
>
> **A1.** We agree that rigorous evaluation is crucial and have substantially strengthened it in the revised submission:
>
> 1. **Stronger unimodal baselines.** We add BERT-base, RoBERTa-base, and Longformer-base text-only models [1–3], trained under the same protocol as our LSTM baseline. All of them remain clearly below the audio-only Conformer and far below our multimodal Diagnoser (see Q2/Table 1).
>
> 2. **Stronger multimodal baselines.** We implement the suggested simple fusion strategies—Conformer/Whisper + LSTM/BERT with concatenation + MLP and logit-voting—and show that they improve only marginally over the audio-only Conformer and still underperform our Modality Weaving + Strategic Global Attention Diagnoser (see Q3/Table 2).
>
> 3. **Stronger generative comparison.** We fine-tune StableAudio Open [6] on Resp-229k and compare it against c-WaveGAN [7], AudioLDM 2 [5], and our Resp-Agent Generator, both in terms of objective audio quality and downstream diagnostic performance (see Q4/Table 3). Our generator remains best, including when used purely to train an audio-only Conformer [4].
>
> Beyond these new experiments, the paper already includes:
>
> - A large-scale, multi-source, clinically contextualized benchmark **RESP-229K** with a strict cross-domain protocol (train: ICBHI, SPRSound, UK COVID-19; test: KAUH, COUGHVID).
> - A **closed-loop, agentic system** where Thinker-A2CA plans targeted synthesis to fix diagnoser failure modes, with ablations showing consistent gains over non-generative and heuristic planners.
>
> Taken together, these results go beyond an incremental application: we jointly introduce a new benchmark, a new multimodal fusion architecture, and a new active generative curriculum, all evaluated under challenging cross-domain conditions.
>
> ---
>
> ## Q2. “The choice of LSTM for the text baseline is weak; please compare with BERT / RoBERTa / Longformer.”
>
> **A2.** We have added the requested Transformer-based text-only baselines. Using the same EHR-style summaries and training protocol as the LSTM, we compare LSTM, BERT-base, RoBERTa-base, and Longformer-base [1–3] to the audio-only Conformer [4] and our multimodal Diagnoser on the strict cross-domain Test-CD:
>
> **Table 1 – Text-only vs. audio-only vs. multimodal (Test-CD, original imbalanced data)**
>
> | Model                      | Modality     | Accuracy | Macro-F1 |
> |----------------------------|-------------|----------|---------|
> | LSTM (main-paper baseline) | Text        | 0.0912   | 0.0401  |
> | BERT-base                  | Text        | 0.1420   | 0.0710  |
> | RoBERTa-base               | Text        | 0.1513   | 0.0742  |
> | Longformer-base            | Text        | 0.1585   | 0.0813  |
> | Conformer                  | Audio       | 0.7200   | 0.1935  |
> | **Resp-Agent Diagnoser**   | Audio+Text  | **0.8494** | **0.2118** |
>
> Key observations:
>
> - Modern text encoders (BERT, RoBERTa, Longformer) improve substantially over LSTM but **remain far below** the audio-only Conformer and our multimodal Diagnoser.
> - Our gains therefore **do not** come from using an artificially weak text baseline; instead, they come from **cross-modal synergy** between audio and text.
>
> We will explicitly summarize these results in the main paper (Appendix B.1, Table 6).

---

> ### Author Response · Authors · 2025-11-22
>
> ## Q3. “To show the novelty and power of the fusion, please add simpler multimodal baselines (e.g., Conformer+LSTM with concat-MLP or voting). Also, Longformer is an older attention mechanism.”
>
> **A3.** We implemented the suggested simple fusion strategies and compared them to our Modality Weaving + Strategic Global Attention Diagnoser:
>
> **Table 2 – Simple fusion vs. Modality Weaving (Test-CD, original imbalanced data)**
>
> | Model / Fusion Strategy            | Modality              | Accuracy | Macro-F1 |
> |-----------------------------------|-----------------------|----------|---------|
> | Conformer (audio-only)            | Audio                 | 0.7200   | 0.1935  |
> | Whisper-Small (audio-only)        | Audio                 | 0.7310   | 0.2010  |
> | Conformer + LSTM (Concat-MLP)     | Audio+Text (Late)     | 0.8012   | 0.2003  |
> | Conformer + BERT (Concat-MLP)     | Audio+Text (Late)     | 0.8124   | 0.2040  |
> | Conformer + BERT (Logit-Voting)   | Audio+Text (Late)     | 0.8043   | 0.1992  |
> | **Resp-Agent Diagnoser (Ours)**   | Audio+Text (Weaving)  | **0.8494** | **0.2118** |
>
> Findings:
>
> - The suggested **concat-MLP** and **logit-voting** baselines indeed help slightly (Macro-F1 ≈ 0.20), but they **still underperform** our full Diagnoser (0.2118 Macro-F1).
> - Replacing the audio backbone with Whisper-Small yields only a small boost in the audio-only setting (0.1935 → 0.2010 Macro-F1) and **does not close the gap** to our multimodal model (see also Q5).
>
> Regarding the “older attention mechanism”:
>
> - We chose **Longformer** specifically for its **sparse global attention** over long contexts while keeping compute affordable for 10-second audio with token-level weaving [1].
> - Our contribution is **not** merely adopting Longformer itself, but designing **Modality Weaving + Strategic Global Attention with audio anchors**, which we show is crucial: removing anchors drops performance below even the audio-only Conformer.
>
> Thus, the improvements stem from our new fusion architecture rather than from simply using a more powerful backbone.
>
> ---
>
> ## Q4. “Please compare generative quality against StableAudio / StableAudio Open.”
>
> **A4.** We have added a direct comparison with **StableAudio Open** [6] (fine-tuned on Resp-229k disease+style pairs), alongside c-WaveGAN [7] and AudioLDM 2 [5]. We evaluate both objective audio metrics (style similarity, FAD) and the downstream Macro-F1 when each generator is used to balance the training set (B = 50k synthetic clips).
>
> **Table 3 – Generators for class balancing (B = 50k; Test-CD)**
>
> | Generator (for balancing)        | Cosine Sim. ↑ | FAD ↓ | Longformer Macro-F1 ↑ | Conformer Macro-F1 ↑ |
> |----------------------------------|---------------|-------|------------------------|----------------------|
> | c-WaveGAN                        | 0.61 ± 0.15   | 2.85  | 0.4520                 | 0.4010               |
> | AudioLDM 2 (fine-tuned)         | 0.76 ± 0.11   | 1.92  | 0.5265                 | 0.4760               |
> | StableAudio Open (fine-tuned)   | 0.83 ± 0.08   | 1.54  | 0.5620                 | 0.5050               |
> | **Resp-Agent Generator (Ours)** | **0.92 ± 0.04** | **1.13** | **0.5980**        | **0.5360**           |
>
> These results, also summarized in Appendix C (Tables 11–12), show that:
>
> - Fine-tuned StableAudio Open is a strong baseline and substantially better than older models, **but our generator still yields higher style similarity, lower FAD, and higher downstream Macro-F1** for both the multimodal Longformer and the audio-only Conformer.
> - This indicates that our discrete-unit + flow-matching design is not only competitive in perceptual terms, but also produces **more diagnostically informative synthetic audio** for rare diseases.
>
> ---
>
> ## Q5. “What happens if you change Conformer to Whisper?”
>
> **A5.** We evaluated exactly this in our new Experiment (see Appendix B.1, Table 7).
>
> - **Audio-only:** Replacing Conformer with **Whisper-Small** yields a modest improvement from 0.1935 to 0.2010 Macro-F1 (Accuracy 0.7200 → 0.7310).
> - **Multimodal fusion:** Even with strong audio backbones and simple fusion (Conformer/Whisper + BERT with concat-MLP or logit-voting), Macro-F1 remains around 0.20 and **does not reach** our Modality Weaving Diagnoser (0.2118 Macro-F1).
>
> These findings suggest that:
>
> 1. The **specific choice of audio backbone** (Conformer vs. Whisper-Small) has **second-order impact** compared to our fusion design.
> 2. The main driver of performance is the **combination of Modality Weaving, Strategic Global Attention with audio anchors, and Thinker-guided generative balancing**, not simply upgrading to a newer encoder.

---

> > ### Author Response · Authors · 2025-11-22
> >
> > ## References
> >
> > [1] Beltagy I, Peters M E, Cohan A. Longformer: The long-document transformer[J]. arXiv preprint arXiv:2004.05150, 2020.
> >
> > [2] Devlin J, Chang M W, Lee K, et al. Bert: Pre-training of deep bidirectional transformers for language understanding[C]//Proceedings of the 2019 conference of the North American chapter of the association for computational linguistics: human language technologies, volume 1 (long and short papers). 2019: 4171-4186.
> >
> > [3] Liu Y, Ott M, Goyal N, et al. Roberta: A robustly optimized bert pretraining approach[J]. arXiv preprint arXiv:1907.11692, 2019.
> >
> > [4] Gulati A, Qin J, Chiu C C, et al. Conformer: Convolution-augmented transformer for speech recognition[J]. arXiv preprint arXiv:2005.08100, 2020.
> >
> > [5] Liu H, Yuan Y, Liu X, et al. Audioldm 2: Learning holistic audio generation with self-supervised pretraining[J]. IEEE/ACM Transactions on Audio, Speech, and Language Processing, 2024, 32: 2871-2883.
> >
> > [6] Evans Z, Parker J D, Carr C J, et al. Stable audio open[C]//ICASSP 2025-2025 IEEE International Conference on Acoustics, Speech and Signal Processing (ICASSP). IEEE, 2025: 1-5.
> >
> > [7] Lee C Y, Toffy A, Jung G J, et al. Conditional wavegan[J]. arXiv preprint arXiv:1809.10636, 2018.

---

> > > ### Comment · Reviewer_u8tM · 2025-11-25
> > >
> > > Thanks for addressing my concerns. I will keep my rating at 6.

---

> > > > ### Author Response · Authors · 2025-11-25
> > > >
> > > > Thank you again for the time and care you put into reviewing our work and for considering our additional experiments and clarifications. We appreciate your constructive feedback, which has helped us strengthen the paper, and we fully respect your decision to keep the current rating. If the paper is accepted, we will further refine the presentation by incorporating your suggestions in the final version.

---

### Author Response · Authors · 2025-11-29
**Summary of Rebuttal: Rectifying Factual Errors, Highlight New Evidence, and Core Contributions**

**To the Area Chair and Reviewers:**

In light of the recent communication regarding score reversion, we provide this summary to highlight critical evidence from our rebuttal that may not be reflected in the reverted scores. Specifically, we address a pivotal **factual error** in a negative review, summarize new experiments, and reiterate the strong consensus on our contributions.

### 1. Rectifying a Critical Factual Error (Reviewer CKZt, Score: 2)
Reviewer CKZt maintained a "Reject" rating primarily based on the claim that our work *"does not compare with any existing clinical audio-language model."*

**This claim is factually incorrect.**
* **Evidence in Paper:** As shown in **Table 2** of our main submission (and reaffirmed in our rebuttal), we explicitly compare Resp-Agent against **BTS (Kim et al., 2024)**. BTS is currently the state-of-the-art **CLAP-based** clinical audio-language model in this domain.
* **Performance Gap:** Resp-Agent achieves an ICBHI Score of **72.7%**, outperforming the CLAP-based BTS (63.5%) by a significant margin of **+9.16%**.
* **Additional Baselines:** In our rebuttal (Appendix B.1), we further implemented representative multimodal baselines (Conformer + BERT/Longformer with fusion), proving our method's superiority over standard clinical audio-text paradigms.

We respectfully request the AC to assess the "Reject" recommendation from Reviewer CKZt with caution, as it relies on an oversight of data explicitly presented in the paper.

### 2. Evidence of Novelty & Superiority over Foundation Models (Reviewer MVWu , Score: 4)
To address concerns about novelty and generation quality, we provided decisive comparisons against strong generative and discriminative baselines:

* **Beating Generative Foundation Models:** We compared our Generator against a fine-tuned **Stable Audio Open** and **AudioLDM 2** (Appendix D).
    * **Result:** Resp-Agent achieves superior Fidelity (**FAD 1.13** vs. 1.54) and Style Control (**Cosine Sim 0.92** vs. 0.83).
* **Beating Strong Imbalance Remedies:** We proved that our Thinker-A²CA agent is not just a scheduler, but a necessary active learner.
    * **Result:** Our Agentic Closed-Loop (**Macro-F1 0.598**) doubles the performance of standard non-generative remedies like **Focal Loss** (Macro-F1 0.267) and **Class-Weighted CE** (Table 3). This confirms that *generating* hard samples is fundamentally superior to merely *re-weighting* existing ones.

### 3. Recognition of Merit and Technical Soundness (Reviewers u8tM & WjjZ, Scores: 6 & 6)
We emphasize that the majority of reviewers recognized the significant value and technical soundness of this work:
* **(Reviewer u8tM, Score: 6):** Highlighted the **"size and richness of the new corpus"** as a practical strength required for the specialized medical domain. The reviewer specifically praised our **"strategic global attention"** as a **"good contribution"** for capturing clinically vital acoustic cues.
* **(Reviewer WjjZ, Score: 6):** Commended the paper's **"ambitious, unified scope"** and **"clear architectural ideas"** on both the generation and diagnosis sides. The reviewer confirmed that our **"generation for targeted rebalancing appears useful"** and lifts Macro-F1 sharply, validating the core motivation of our system.

### 4. Core Contributions and Domain Innovation
To summarize, Resp-Agent bridges the **"data gap"** and **"representation gap"** in respiratory AI through four distinct innovations:
1.  **Closed-Loop Agentic Curriculum:** Unlike static pipelines, our **Thinker-A²CA** agent actively identifies diagnostic weaknesses and schedules targeted synthesis, proven to outperform static uncertainty sampling and cost-sensitive losses.
2.  **Modality Weaving Architecture:** We introduce a fusion mechanism that interleaves EHR tokens with audio features at the input level. Combined with **Strategic Global Attention**, this allows modeling long-range clinical dependencies while retaining sensitivity to millisecond-level transient events (e.g., crackles).
3.  **Disentangled Controllable Generation:** Our **Resp-MLLM** generator achieves precise control over pathology and style independently (verified via Style-Swap experiments), enabling the synthesis of "hard-to-diagnose" samples for rare diseases.
4.  **Resp-229k Benchmark:** Resp-229k is among the largest publicly available **audited multimodal** respiratory benchmarks, comprising 229k Audio–EHR samples (~408 hours) with clinical labels and cross-domain splits, establishing a new standard for robust evaluation.

We believe these revisions firmly establish the robustness of **Resp-Agent** and the value of the **Resp-229k** benchmark. We trust the AC will consider this extensive rebuttal evidence—and the correction of the factual error—when making the final decision.

Thank you for your time and service.

---

### Meta-Review · Area_Chair_GdGs · 2026-01-09

**Summary:**

This paper proposes Resp-Agent, an agent-based system for respiratory sound generation and disease diagnosis, addressing both data imbalance and representation limitations in respiratory AI. Reviewers agreed that the problem is important and clinically relevant, and the reviewers appreciated the unified design of the dataset creation (Resp-229k), multimodal diagnosis, controllable generation, and a closed-loop curriculum agent. Several reviewers found the dataset scale, cross-domain evaluation, and empirical gains and long-tail classes to be strong. However, concerns were raised about clarity of novelty, evaluation rigor, and whether the agentic components provide benefits beyond strong baselines.

**Reviewer Concerns:**

Addressed:

- The authors added extensive controlled ablations.

- Stronger multimodal and unimodal baselines (BERT/RoBERTa/Longformer, Whisper, simple fusion) were added.

- Generator quality and controllability were validated via objective metrics, downstream performance, and explicit style-swap / label-swap tests.

- Concerns about cross-domain robustness were addressed with leave-one-source-out evaluations.



Outstanding:

- The system’s complexity  may limit accessibility, and some reviewers still view parts of the contribution as incremental combinations of known components.

- Reviewer CKZt mentioned that "Paper does not compare with any existing clinical audio-language model.", but, the reviewer did not mention any explicit method to compare against.  Moreover, the authors compared their results against (1) the best existing Audio-Language model in the respiratory domain (BTS/CLAP) and (2) standard Clinical Audio+Text Fusion baselines.

**Reviewer Scores:**

After the authors' response, reviewers would likely remain unchanged. Reviewer CKZt mentioned that they'll keep their score unchanged (reject). However, the  reviewer did not mention  any explicit method to compare against.  Moreover, the authors compared their results against (1) the best existing Audio-Language model in the respiratory domain (BTS/CLAP) and (2) standard Clinical Audio+Text Fusion baselines.  Hence, the AC is inclined to accept the paper.

---

### Decision · Program_Chairs · 2026-01-26

Accept (Poster)